# Lysosomal protein surface expression discriminates fat- from bone-forming human mesenchymal precursor cells

Jiajia Xu[1†], Yiyun Wang[1†], Ching-Yun Hsu[1], Stefano Negri[1], Robert J Tower[1,2], Yongxing Gao[1], Ye Tian[1,3], Takashi Sono[1], Carolyn A Meyers[1], Winters R Hardy[1,4], Leslie Chang[1], Shuaishuai Hu[4], Nusrat Kahn[4], Kristen Broderick[5], Bruno Péault[4,6*], Aaron W James[1,4*]

[1]Departments of Pathology, Johns Hopkins University, Baltimore, United States; [2]Departments of Orthopaedics, Johns Hopkins University, Baltimore, United States; [3]Department of Oral and Maxillofacial Surgery, School of Stomatology, China Medical University, Shenyang, China; [4]UCLA and Orthopaedic Hospital Department of Orthopaedic Surgery and the Orthopaedic Hospital Research Center, Los Angeles, United States; [5]Departments of Plastic Surgery, Johns Hopkins University, Baltimore, United States; [6]Center For Cardiovascular Science and Center for Regenerative Medicine, University of Edinburgh, Edinburgh, United Kingdom

*For correspondence:
BPeault@mednet.ucla.edu (BP);
awjames@jhmi.edu (AWJ)

[†]These authors contributed equally to this work

**Abstract** Tissue resident mesenchymal stem/stromal cells (MSCs) occupy perivascular spaces. Profiling human adipose perivascular mesenchyme with antibody arrays identified 16 novel surface antigens, including endolysosomal protein CD107a. Surface CD107a expression segregates MSCs into functionally distinct subsets. In culture, CD107a[low] cells demonstrate high colony formation, osteoprogenitor cell frequency, and osteogenic potential. Conversely, CD107a[high] cells include almost exclusively adipocyte progenitor cells. Accordingly, human CD107a[low] cells drove dramatic bone formation after intramuscular transplantation in mice, and induced spine fusion in rats, whereas CD107a[high] cells did not. CD107a protein trafficking to the cell surface is associated with exocytosis during early adipogenic differentiation. RNA sequencing also suggested that CD107a[low] cells are precursors of CD107a[high] cells. These results document the molecular and functional diversity of perivascular regenerative cells, and show that relocation to cell surface of a lysosomal protein marks the transition from osteo- to adipogenic potential in native human MSCs, a population of substantial therapeutic interest.

## Introduction

Within mammalian white adipose tissue (WAT), a perivascular population of mesenchymal progenitor cells is well documented with respect to multipotency and tissue renewal capabilities (*Corselli et al., 2012*; *Kramann et al., 2016*). The ability of human WAT-resident perivascular cells to differentiate into bone-forming osteoblasts and incite or participate in bone repair is also well known (*Askarinam et al., 2013*; *Chung et al., 2014*; *James et al., 2012a*; *James et al., 2012b*; *James et al., 2012c*; *Lee et al., 2015*; *Meyers et al., 2018b*; *Tawonsawatruk et al., 2016*) (see (*James et al., 2017*; *James and Péault, 2019*) for reviews). The bulk of WAT-resident perivascular cells with mesenchymal progenitor cell attributes reside in the *tunica adventitia* – the outer collagen-rich sheath of blood vessels (*Corselli et al., 2012*; *James et al., 2012a*; *West et al., 2016*). Micro-vascular pericytes, although less frequent in absolute numbers, also demonstrate progenitor cell attributes (*Chen et al., 2013*; *Crisan et al., 2009*; *Crisan et al., 2008*). With several recent studies

from our group in human (*Ding et al., 2019*; *Hardy et al., 2017*) and mouse WAT (*Wang et al., 2020*), it is clear that perivascular cells, including those found within the *tunica adventitia* (adventitial cells or adventicytes), demonstrate more phenotypic and functional diversity than previously understood.

CD107a (lysosome-associated membrane protein-1, LAMP1) is a member of a family of structurally related type one membrane proteins predominantly expressed in lysosomes and other intracellular vesicles (*Carlsson et al., 1988*; *de Saint-Vis et al., 1998*; *Defays et al., 2011*; *Ramprasad et al., 1996*). CD107a is far less frequently expressed on the cell surface, which is the result of both trafficking of nascent protein to the plasma membrane as well as the fusion of late endosomes and lysosomes to the cell membrane (*Akasaki et al., 1993*; *Dell'Angelica et al., 2000*). In inflammatory cells, surface CD107a reflects the state of activation (*Janvier and Bonifacino, 2005*) and has been implicated in cell adhesion (*Kannan et al., 1996*; *Min et al., 2013*). In separate reports, CD107a has been described in intracellular vesicles in both osteoblasts and adipocytes (*Bandeira et al., 2018*; *Solberg et al., 2015*), yet beyond this, essentially nothing is known regarding CD107a in mesenchymal stem cell fate or differentiation decisions.

Here, antibody array screening of FACS-defined stromal vascular fraction (SVF) perivascular cells identified several novel cell surface antigens, including CD107a, enriched within subpopulations of human adventicytes and pericytes. Flow cytometry and immunohistochemical analyses confirmed that cells with membranous surface CD107a expression reside in a perivascular microanatomical niche within WAT. CD107a$^{high}$ cells represent an adipocyte precursor cell, while CD107a$^{low}$ cells represent progenitors with increased osteoblast potential. CD107a trafficking to the cell surface was observed to occur during early adipocyte differentiation – results confirmed by single-cell RNA sequencing datasets from mouse and human adipose tissues. Upon transplantation into immunocompromised rodents, CD107a$^{low}$ cells robustly induce bone formation, both in an intramuscular ectopic ossicle assay in mice and a lumbar spine fusion rat model. These results suggest that cell surface CD107a divides osteoblast from adipocyte perivascular precursors within human tissues.

## Results

### Identification of CD107a as a novel cell surface antigen expressed among adipose tissue (AT)-resident perivascular stem cells

To identify new markers that may define subsets of perivascular cells, a cell surface antigen screen (Lyoplate) was performed on previously defined perivascular cell fractions (*Crisan et al., 2008*; *James et al., 2012c*; *Xu et al., 2019*), including CD34$^+$CD146$^-$ adventitial cells and CD146$^+$CD34$^-$ pericytes after exclusion of non-viable, endothelial, and hematopoietic cells (PI$^+$ CD31$^+$ or CD45$^+$ fractions) (*Table 1*). Several markers were confirmed to be highly expressed among both adventicytes and pericytes, including, for example, the progenitor cell and MSC marker CD90 (Thy-1) and the perivascular cell antigen CD140b (PDGFRβ). Novel markers to divide perivascular progenitors ranged broadly, including the endolysosomal protein CD107a (32% and 82% expression among adventicytes and pericytes, respectively). Another endolysosomal protein, CD107b, was also present on each perivascular cell fraction (13% and 46% expressing adventicytes and pericytes, respectively). Other markers noted to be expressed differentially in subsets of adventicytes and pericytes included CD98, CD140a (PDGFRα), CD142, CD165, CD200, and CD271 (NGFR) (*Table 1*).

Next, previously derived transcriptomics data on human perivascular cells were analyzed to confirm *LAMP1* gene expression, encoding CD107a. Using WAT-derived pericytes (n = 3 samples, GEO dataset: GSE125545) or adventitial perivascular stem cells (n = 3 samples, GEO dataset: GSE130086) (*Xu et al., 2019*), high expression of the *LAMP1* gene was confirmed (mean FPKM values of 9.576 and 9.619, respectively).

Spatial localization of CD107a was next assessed by immunostaining of subcutaneous WAT (*Figure 1A–E*, n = 3 samples). CD107a immunoreactivity was found most frequently within the outermost layers of larger arteries (*Figure 1B*) and veins (*Figure 1C*). Within arteries, the outer *tunica adventitia* showed a high frequency of CD107a$^+$CD34$^+$ cells (*Figure 1B1*), while the inner *adventitia* showed predominantly CD107a$^-$CD34$^+$ cells (*Figure 1B2*). The smooth muscle media largely did not show CD107a immunoreactivity (*Figure 1B3*), which was confirmed by dual immunohistochemistry for CD107a and αSMA (*Figure 1—figure supplement 1A*). Co-expression with the recently

**Table 1.** Surface antigens expressed within human adventitial cells versus pericytes.
Results derived from Lyoplate analysis of CD34$^+$CD146$^-$CD45$^-$CD31$^-$ adventitial cells or CD146$^+$CD34$^-$CD45$^-$CD31$^-$ pericytes.

| CD marker | Protein name | Frequency in adventitial cells (CD34$^+$CD146$^-$CD45$^-$CD31$^-$) | Frequency in pericytes (CD146$^+$CD34$^-$CD45$^-$CD31$^-$) |
|---|---|---|---|
| CD90 | Thy-1 | 97% | 70% |
| CD91 | Low-density lipoprotein-related receptor | 97% | 61% |
| CD95 | Fas receptor (TNFRSF6) | 42% | 22% |
| CD98 | Large neutral amino acid transporter (LAT1) | 17% | 65% |
| CD105 | Endoglin | 47% | 14% |
| CD107a | Lysosomal-associated membrane protein 1 (LAMP1) | 32% | 82% |
| CD107b | Lysosomal-associated membrane protein 2 (LAMP2) | 13% | 46% |
| CD130 | Interleukin six beta transmembrane protein | 39% | 61% |
| CD140a | Platelet-derived growth factor receptor alpha (PDGFRA) | 82% | 13% |
| CD140b | Platelet-derived growth factor receptor beta (PDGFRB) | 97% | 34% |
| CD142 | Tissue factor, PTF, Factor III, or thromboplastin | 47% | 75% |
| CD147 | Basigin (BSG) | 99% | 99% |
| CD151 | Raph blood group | 71% | 100% |
| CD164 | Sialomucin core protein 24 or endolyn | 91% | 97% |
| CD165 | AD2 | 77% | 21% |
| CD271 | Nerve growth factor receptor (NGFR) | 64% | 10% |

described adventitial marker Gli1 was assessed (*Kramann et al., 2016*), which showed little overlap with CD107a immunoreactive adventitial cells (*Figure 1—figure supplement 1B*). Smaller caliber arteries (*Figure 1D*) and veins showed a high frequency of dual expressing CD107a$^+$CD34$^+$ cells within the *adventitia*. Capillaries within WAT showed some CD107a immunoreactive pericytes, which co-expressed CD146 but not CD31, and were present in an abluminal location (*Figure 1E*, appearing yellow). CD107a immunoreactivity within the perivascular mesenchymal niche was confirmed across other AT depots, including pericardial, perigonadal, perirenal, and omental human fat (*Figure 1—figure supplement 2*, n = 3 samples per depot). Small and medium caliber vessels showed perivascular immunoreactivity across all adipose depots.

Next, flow cytometry demonstrated a spectrum of CD107a membranous staining across the viable, non-endothelial/noninflammatory cells of human WAT (*Figure 1F*). The PI$^-$CD31$^-$CD45$^-$ component of SVF was divided by FACS into CD107a$^{low}$ and CD107a$^{high}$ cell populations for further analysis (*Figure 1F,G*). Mean frequency of CD107a$^{low}$ mesenchymal cells represented 33.75% of viable SVF, while mean frequency of CD107a$^{high}$ mesenchymal cells represented 5.20% (*Supplementary file 1*). Flow cytometry analysis was next performed within CD107a$^{high}$CD31$^-$CD45$^-$ and CD107a$^{low}$CD31$^-$CD45$^-$ mesenchymal populations (*Figure 1H,I*). High expression of CD107a was confirmed by flow cytometry among freshly sorted CD107a$^{high}$ cell preparations (*Figure 1I*). Concordant with histological observations, CD34$^+$ and CD146$^+$ cells were identified in both CD107a$^{low}$ and CD107a$^{high}$ cell fractions (*Figure 1H,I*, mean frequencies reported in *Supplementary file 2, 3*). These data confirmed that surface CD107a expression is present in both perivascular native MSC niches within WAT, and that surface CD107a could be used to prospectively purify mesenchymal cell subpopulations with disparate staining intensities.

Next, canonical markers of culture-expanded human MSCs were examined by flow cytometry in freshly sorted CD107a$^{low}$ and CD107a$^{high}$ cells, including CD44, CD73, CD90, and CD105 (*Supplementary file 4*, *Figure 1—figure supplement 3*). With the exception of CD105, all markers showed overall similar expression patterns across CD107a$^{low}$ and CD107a$^{high}$ cell populations (n = 3 samples per group). CD105 expression was disproportionately present within CD107a$^{high}$ mesenchymal cells (mean frequency 0.44% and 8.12% among CD107a$^{low}$ and CD107a$^{high}$ cells, respectively).

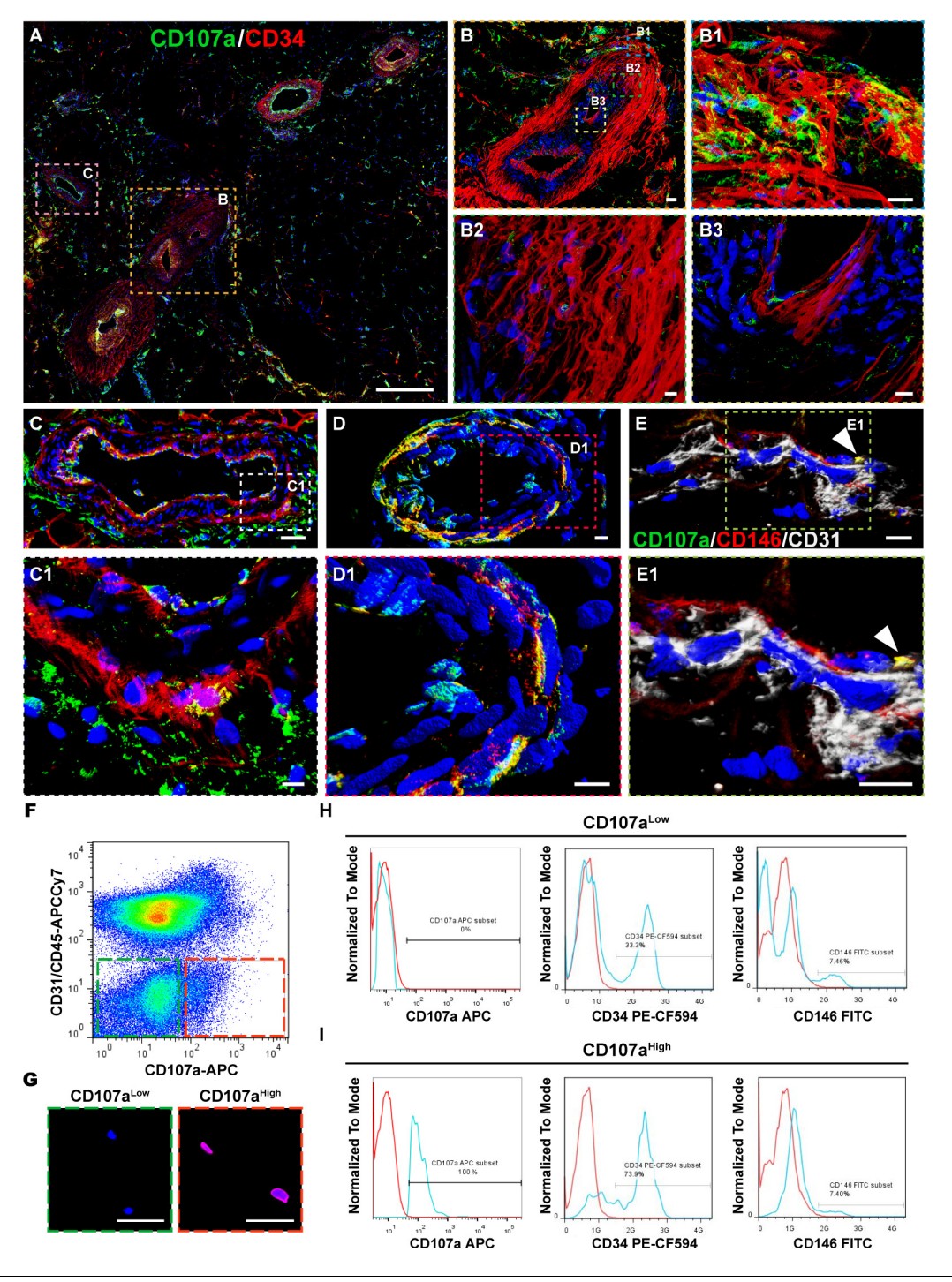

**Figure 1.** Perivascular CD107a expression typifies a subset of perivascular cells within human subcutaneous white adipose tissue (WAT). Immunofluorescent staining of CD107a (green) and CD34 (red) in human adipose tissue. (**A**) Tile scan. (**B**) Larger artery in cross-section, including (**B1**) outer tunica adventitia, (**B2**) inner tunica adventitia, and (**B3**) tunica media and intima. (**C**) Larger vein in cross-section, including (**C1**) high magnification of vessel wall. (**D**) Smaller caliber artery in cross-section, including (**D1**) high magnification of vessel wall. (**E**) Capillary in longitudinal cross-section, and (**E1**) high magnification. (**F**) Representative FlowJo plot to demonstrate partitioning CD107a^low^CD31^-^CD45^- and CD107a^high^CD31^-^CD45^- fractions from human stromal vascular fraction (SVF). Frequency of CD107a^low/high^ cells across samples is shown in ***Supplementary file 1*** (N = 8 samples). (**G**) Confirmatory immunofluorescent staining of FAC-sorted CD107a^low^ and CD107a^high^ mesenchymal cells

*Figure 1 continued on next page*

*Figure 1 continued*

(CD31⁻CD45⁻PI⁻ cells). (**H,I**) Representative flow cytometry analysis of freshly isolated CD107a^low and CD107a^high mesenchymal cells, including CD107a, CD34, and CD146. Frequency of expression is shown in relation to isotype control (blue vs. red lines). Frequency of CD34⁺ and CD146⁺ cells across all samples is shown in *Supplementary file 2*, *3* (N = 4 samples). Scale bars: 500 µm (A), 50 µm (B,C,G) and 10 µm (B1–B3,C1,D,D1,E,E1). The online version of this article includes the following figure supplement(s) for figure 1:

**Figure supplement 1.** Comparison of CD107a expression with either αSMA or Gli1 in human adipose tissue.
**Figure supplement 2.** Perivascular CD107a immunohistochemical staining in diverse human adipose tissues.
**Figure supplement 3.** Flow cytometry of canonical human mesenchymal stem cell (MSC) markers among freshly isolated CD107a^low/high cells from a representative patient sample.

## CD107a^low AT-derived stromal cells represent osteoblast precursor cells

CD107a^low and CD107a^high cells were again derived from the CD31⁻CD45⁻ fraction of adipose tissue samples, and in vitro properties examined (*Figure 2*). Morphology of adherent CD107a^low and CD107a^high cells was broadly similar, with a fibroblastic shape (*Figure 2—figure supplement 1*). CD107a^low cells demonstrated a higher proliferative rate in comparison to CD107a^high cells (*Figure 2A*). Among freshly isolated cells, the vast majority of colony forming units-fibroblast (CFU-F) was identified within the CD107a^low cell fraction (*Figure 2B,C*). Among equivalent cells at passage 4, an enrichment in CFU-F was still observed in CD107a^low cells (*Figure 2D*). CFUs-osteoblast (CFU-OB) likewise showed a similar enrichment among CD107a^low cells. CFU-OB assays performed in growth medium showed alkaline phosphatase (ALP)⁺ colonies among CD107a^low cells only (*Figure 2E,F*). The same experiment performed in osteogenic differentiation medium showed an enrichment in CFU-OB among CD107a^low cells (*Figure 2G,H*). Among passaged cells in sub-confluent monolayer, osteogenic differentiation was next examined (*Figure 2I–O*). ALP staining and quantification demonstrated an enrichment among CD107a^low cells (*Figure 2I,J*). Bone nodule formation was likewise increased in CD107a^low cells as compared to CD107a^high counterparts (*Figure 2K,L*). Osteogenic gene expression across timepoints of differentiation likewise showed an enrichment for *RUNX2 (Runt related transcription factor 2)*, *ALPL*, and *osteopontin (SPP1)* (*Figure 2M–O*). Thus, the CD107a^low mesenchymal component of human WAT contains a precursor cell population with high osteoblastogenic potential.

## CD107a^high AT-derived stromal cells represent adipocyte precursor cells

Converse experiments to assay adipogenesis were next performed among cell subsets with differential CD107a expression (*Figure 3*). Among freshly isolated cells, essentially all CFU-adipocyte (CFU-AD) were found within the CD107a^high cell population (*Figure 3A,B*). Next, sub-confluent CD107a^low and CD107a^high cells were propagated under adipogenic differentiation conditions (*Figure 3C–G*). Oil red O staining was significantly more abundant among the CD107a^high cells (*Figure 3C,D*). Adipogenic gene expression was next assessed along adipogenic differentiation (*Figure 3E–G*). All marker gene transcripts showed significant enrichment among CD107a^high cells in comparison to CD107a^low counterparts, including *peroxisome proliferator-activated receptor gamma (PPARG)*, *lipoprotein lipase (LPL)*, and *fatty acid-binding protein 4 (FABP4)*.

Finally, to confirm that CD107a^low and CD107a^high mesenchymal cell fractions both represented multipotent precursor cells, parallel chondrogenic differentiation assays were performed in three dimensional micromass culture (*Figure 3—figure supplement 1*). Here, both cell populations demonstrated a progressive increase in cartilage associated gene expression after 7 d in chondrogenic differentiation conditions. No significant differences in cartilage gene expression were observed between CD107a^low and CD107a^high cell fractions (*Figure 3—figure supplement 1A*). Alcian blue stained sections of micromass cultures at 21 d of differentiation likewise showed a similar appearance between CD107a^low and CD107a^high cells (*Figure 3—figure supplement 1B*). Thus, CD107a^low and CD107a^high subpopulations of human SVF both house multipotent mesenchymal cells, but with considerably divergent osteoblastic and adipocytic differentiation potentials.

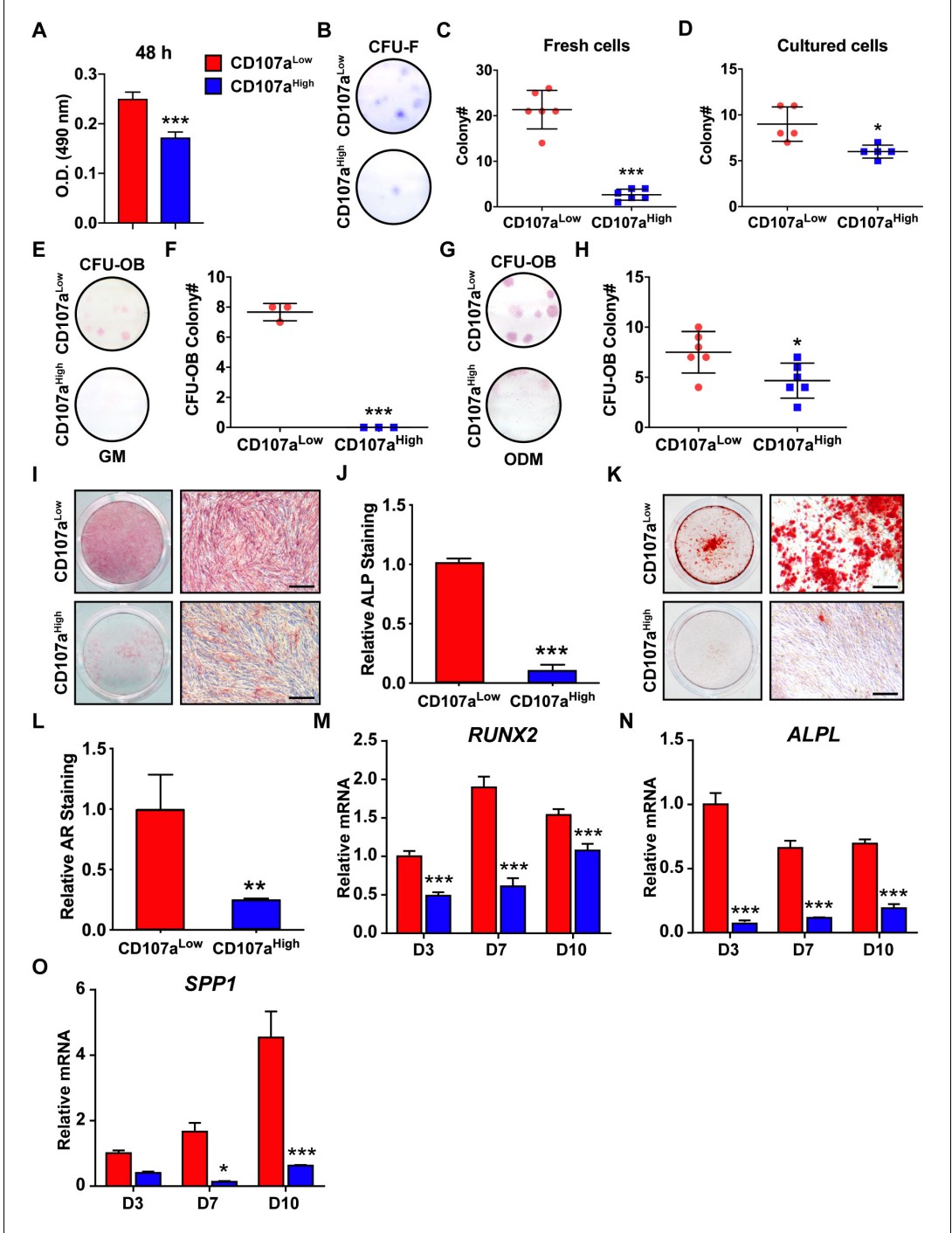

**Figure 2.** Stem/osteoprogenitor cell identity of CD107a$^{low}$ mesenchymal cells. CD107a$^{low}$CD31$^-$CD45$^-$ and CD107a$^{high}$CD31$^-$CD45$^-$ cells derived from the same sample of human subcutaneous WAT were exposed to the indicated growth or osteogenic conditions. (A) Cell proliferation among CD107a$^{low}$ and CD107a$^{high}$ mesenchymal cells, by MTS assays at 48 hr. (B–D) Fibroblastic colony formation frequency (CFU-F) among human CD107a$^{low}$ and CD107a$^{high}$ mesenchymal cells, shown by (B) representative images among freshly isolated cells, (C) CFU-F quantification among freshly isolated cells, (D) CFU-F quantification among passage 4 cells. Whole well images are shown. (E–H) Osteoblastic colony formation frequency (CFU-OB) detected in human CD107a$^{low}$ and CD107a$^{high}$ cells. Experiments performed in growth medium (GM) (E,F) or osteogenic differentiation medium (ODM) (G,H). Whole well images are shown. (I,J) Alkaline phosphatase (ALP) staining and photometric quantification at d 10 of osteogenic differentiation among human CD107a$^{low}$ and CD107a$^{high}$ cells. Representative whole well and high magnification images are shown. (K,L) Alizarin red (AR) staining and photometric quantification at d 10 of osteogenic differentiation among human CD107a$^{low}$ and CD107a$^{high}$ cells. Representative whole well and high magnification images are shown. (M–O) Osteogenic gene expression among human CD107a$^{low}$ and CD107a$^{high}$ cells by qRT-PCR at d 3, 7, and 10 of

*Figure 2 continued on next page*

Figure 2 continued

differentiation, including (**M**) *Runt related transcription factor 2* (*RUNX2*), (**N**) *ALPL*, and (**O**) *Osteopontin* (*SPP1*). Osteogenic differentiation examined in N = 3 human cell preparations, and at least experimental triplicate. Dots in scatterplots represent values from individual wells, while mean and one SD are indicated by crosshairs and whiskers. In column graphs, mean values and one SD are shown. *p<0.05; **p<0.01; ***p<0.01 in relation to corresponding CD107a$^{low}$ cell population. Statistical analysis was performed using a two-tailed Student t-test (**A–L**) or two-way ANOVA followed by Sidak's multiple comparisons test (**M–O**). Experiments performed in at least biologic triplicate. Scale bars: 200 μm.

The online version of this article includes the following figure supplement(s) for figure 2:

**Figure supplement 1.** Representative morphology of confluent human CD107a$^{low}$ and CD107a$^{high}$ cells.

## CD107a traffics to the cell surface during early adipocyte differentiation

Cell surface expression of CD107a results predominantly from trafficking of endolysosomal CD107a$^+$ vesicles to the cell surface. To investigate, unpurified ASCs were exposed to growth conditions or adipogenic differentiation conditions and cell surface CD107a was assessed by immunocytochemistry or flow cytometry (*Figure 3H–J*). After 3 d exposure to adipogenic conditions, a 12.03 fold increase in immunostaining intensity and a 253.5% increase in the number of CD107a$^{high}$ cells were noted. Flow cytometry across several other human cell types confirmed this finding, including FACS-purified perivascular stem cells (PSCs) (*Figure 3K*) and culture-defined human bone marrow mesenchymal stem cells (BMSCs) (*Figure 3L*) (188.1–455.4% increase in CD107a$^{high}$ cell frequency). Parallel experiments were performed under osteogenic differentiation conditions, which found no significant increase in CD107a staining intensity (*Figure 3—figure supplement 2*). The increase in membranous CD107a during early adipogenesis was reversed by vacuolin-1 (Vac-1), an inhibitor of Ca$^{2+}$-dependent fusion of lysosomes with the cell membrane. Treatment with Vac-1 significantly decreased the frequency of CD107a$^{high}$ ASCs, and prevented an increase of cell surface CD107a by adipogenic conditions (*Figure 3M,N*).

In order to confirm that exocytosis is a common feature of early adipogenic differentiation, existing single-cell RNA sequencing datasets of human and mouse AT-derived cells were re-assessed (*Merrick et al., 2019*). Among human AT-derived cells, re-clustering and cell trajectory analysis identified early progenitor cells (expressing *DPP4* and *CD55*), late progenitor cells (expressing *GGT5* and *F3*), as well as an intermediate cell type (middle progenitor cells) (*Figure 3O–Q*). KEGG terms for exocytosis showed enrichment within early and middle progenitor cells, as visualized by heatmaps across pseudotime (*Figure 3R*) and normalized expression of overall exocytosis gene activation (*Figure 3S*). Similar results linking activation of exocytosis to early adipogenic differentiation were obtained from mouse subcutaneous WAT (*Figure 3—figure supplement 3*; *Merrick et al., 2019*). Here, after re-clustering and cell trajectory analysis (*Figure 3—figure supplement 3A–C*), normalized expression of exocytosis gene activation identified similar trends as in human cells (*Figure 3—figure supplement 3D,E*). Normalized exocytosis gene activation scores were again enriched within early *Dpp4*-expressing progenitor cells in comparison to more mature *Dlk1*-expressing pre-adipocytes.

Knockdown (KD) experiments in human ASCs and CD107a$^{low/high}$ cells did not identify a significant functional role for CD107a in osteo/adipogenesis (*Figure 3—figure supplement 4*). SiRNA-mediated KD of *LAMP1* (encoding CD107a) showed no appreciable effect on the osteogenic differentiation of human ASCs (*Figure 3—figure supplement 4A–F*). During adipogenesis, *LAMP1* KD ASCs and CD107a$^{low}$ adventicytes showed a modest increase in lipid droplet accumulation (*Figure 3—figure supplement 4G,K,L,P,Q*) and likewise a modest increase in expression of adipogenic marker genes (*Figure 3—figure supplement 4H–J,M–O*). The high adipogenic differentiation potential of CD107a$^{high}$ cells was not significantly altered with *LAMP1* KD (*Figure 3—figure supplement 4P,Q*). Thus, high expression of membranous CD107a, rather than having vital function in cellular differentiation, correlates with exocytosis during early adipogenic differentiation.

## Transcriptomic analysis suggests a progenitor cell phenotype for CD107a$^{low}$ cells

Differences in differentiation potential were next investigated using transcriptomic analysis of CD107a$^{low}$ and CD107a$^{high}$ stromal cells. RNA sequencing comparative analysis was performed

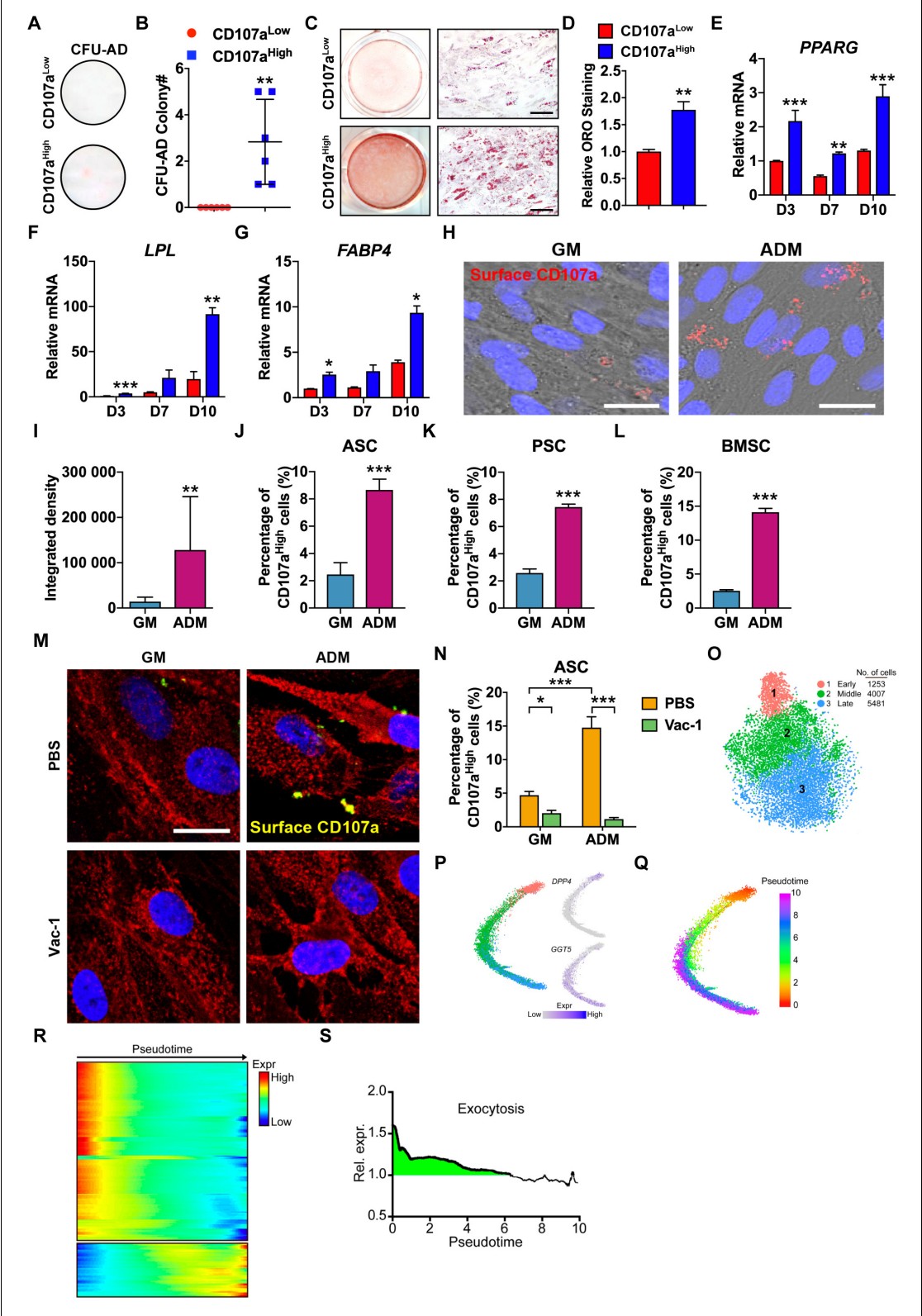

**Figure 3.** Adipoprogenitor cell identity of CD107a[high] human mesenchymal cells and correlation to exocytosis during early adipogenic differentiation. (A–G) CD107a[low]CD31[-]CD45[-] and CD107a[high]CD31[-]CD45[-] cells derived from human subcutaneous WAT were exposed to adipogenic differentiation conditions. (A,B) Adipocyte colony formation frequency (CFU-AD) detected in human CD107a[low] and CD107a[high] cells. Whole well images shown. (C,D) Oil red O (ORO) staining and photometric quantification at d7 of adipogenic differentiation among human CD107a[low] and CD107a[high] cells.

*Figure 3 continued on next page*

*Figure 3 continued*

Representative whole well and high magnification images shown. (E–G) Adipogenic gene expression among human CD107a[low] and CD107a[high] cells by qRT-PCR at d 3, 7, and 10 of differentiation, including (E) *Peroxisome proliferator-activated receptor-γ* (*PPARG*), (F) *Lipoprotein lipase* (*LPL*), and (G) *Fatty acid-binding protein 4* (*FABP4*). (H) Immunocytochemical staining of membranous CD107a in the presence of growth medium (GM) or adipogenic differentiation medium (ADM) after 3 d using human, culture-defined adipose-derived stem cells (ASCs). CD107a immunoreactivity appears red, nuclear counterstain appears blue. (I) Photographic quantification of membranous CD107a immunofluorescence under GM or ADM conditions. (J–L) Induction of membranous CD107a expression after adipogenic differentiation across cell types, including (J) culture-defined human ASCs, (K) FACS-purified human perivascular stem cells (PSC), and (L) culture-defined human BMSCs, assessed by flow cytometry after 3 d under GM or ADM conditions. (M,N) Trafficking of CD107a to the cell surface during adipogenesis was inhibited after treatment with Vacuolin-1 (Vac-1, 1 μM), assessed by CD107a immunostaining (M) and flow cytometry (N) after 3 d under GM and ADM conditions. The cell membrane was labeled using Wheat Germ Agglutinin Conjugates (red), while overlap with CD107a immunostaining appears yellow, and DAPI nuclear counterstain appears blue. (O) Dimensional reduction and unsupervised clustering of human stromal vascular fraction (SVF) adipogenic lineage from subcutaneous WAT revealed three cell groups. (P) Trajectory analyses of human SVF adipogenic lineage, colored based on their unsupervised clustering identity. *DPP4* (early) and *GGT5* (late) expression were used to identify trajectory origin. (Q) Pseudotemporal cell ordering along differentiation trajectories. Pseudotime is depicted from red to purple. (R) Expression heatmap across pseudotime of genes associated with exocytosis. (S) Combined, normalized expression of exocytosis genes shows enrichment (>1) in early progenitors (green shaded area), while more differentiated cells show reduced average expression. Dots in scatterplots represent values from individual wells, while mean and one SD are indicated by crosshairs and whiskers. In column graphs, mean values and one SD are shown. *p<0.05; **p<0.01; ***p<0.001. Statistical analysis was performed using a two-tailed Student t-test (B,D,I–L) or two-way ANOVA followed by Sidak's multiple comparisons test (E–G,N). Experiments performed in at least biologic triplicate. Black scale bar: 100 μm; white scale bar: 20 μm.

The online version of this article includes the following figure supplement(s) for figure 3:

**Figure supplement 1.** Chondrogenic differentiation among human CD107a[low] and CD107a[high] cells.

**Figure supplement 2.** No change in membranous CD107a during osteogenic differentiation among human ASCs.

**Figure supplement 3.** Single-cell RNA sequencing and cell trajectory analysis delineate exocytosis of adipocyte progenitors in mouse subcutaneous white adipose tissue (WAT).

**Figure supplement 4.** Effect of *LAMP1* knockdown (encoding CD107a) on osteogenic and adipogenic differentiation in human ASCs and CD107a[low/high] cells.

across these two stromal cells. Clear separation between gene expression profiles was observed when comparing CD107a[low] with CD107a[high] stromal cells, as assessed by principal component analysis (*Figure 4A*). Further confirming our FACS purification, endothelial and inflammatory marker genes were rarely or not expressed among CD107a[low] and CD107a[high] stromal cells (*Figure 4—figure supplement 1*). Progenitor cell markers were expressed among both CD107a[low] and CD107a[high] stromal cells, with some subtle differences noted (*Figure 4B*). Transcripts of *MYC*, *LEPR* (Leptin receptor), *MCAM* (CD146), and *PDGFRA* (Platelet-derived growth factor receptor α) while expressed across all samples were enriched among CD107a[low] cells. Likewise, *NES* (Nestin), *THY1* (CD90), *PDGFRB* (Platelet-derived growth factor receptor β) and *TBX18* (T-Box transcription factor 18) were expressed across all samples, but more highly among CD107a[high] cells. Other typical MSC markers were more evenly distributed across cell preparations, including *CD44*, and *NT5E* (CD73). Consistent with in vitro differentiation potential, genes associated with adipogenic differentiation were highly expressed among CD107a[high] stromal cells, such as *FABP4* (Fatty acid-binding protein 4), *LPL* (Lipoprotein lipase), *PPARGC1A* (PPARG coactivator 1 α), and *CEBPA* (CCAAT enhancer binding protein α). In addition, negative regulators of adipogenesis were increased among CD107a[low] stromal cells, such as *KLF2* (Krüppel-like factor 2), *KLF3*, *SIRT1* (Sirtuin 1), and *DDIT3* (DNA Damage Inducible Transcript 3) (*Figure 4C*; *Banerjee et al., 2003*; *Pereira et al., 2004*; *Sue et al., 2008*; *Zhou et al., 2015*).

QIAGEN Ingenuity Pathway Analysis (IPA) showed that the activated pathways in CD107a[high] stromal cells are associated with the positive regulation of adipogenesis, including, for example, white adipose tissue browning pathway and Sirtuin signaling (Z scores 1.342 and 1.387; CD107a[high] compared with CD107a[low] stromal cells; *Figure 4D*; *Kurylowicz, 2019*). Conversely, upregulated signaling pathways in CD107a[low] stromal cells included Wnt/β-catenin signaling as well as pathways associated with cellular respiration and metabolism, including Oxidative Phosphorylation and Glutathione metabolism (Z scores −1; CD107a[high] compared with CD107a[low] stromal cells; *Figure 4D* and *Figure 4—figure supplement 2*). In order to further evaluate differences in CD107a cell fractions, pathway analyses were next cross-referenced with prior AT-derived single-cell RNA sequencing data (see again *Figure 3O*). Highly enriched GO terms among CD107a[low] stromal cells were likewise found to be enriched among *DPP4*[+] cell fractions during 'early' pseudotime (*Figure 4E*). This

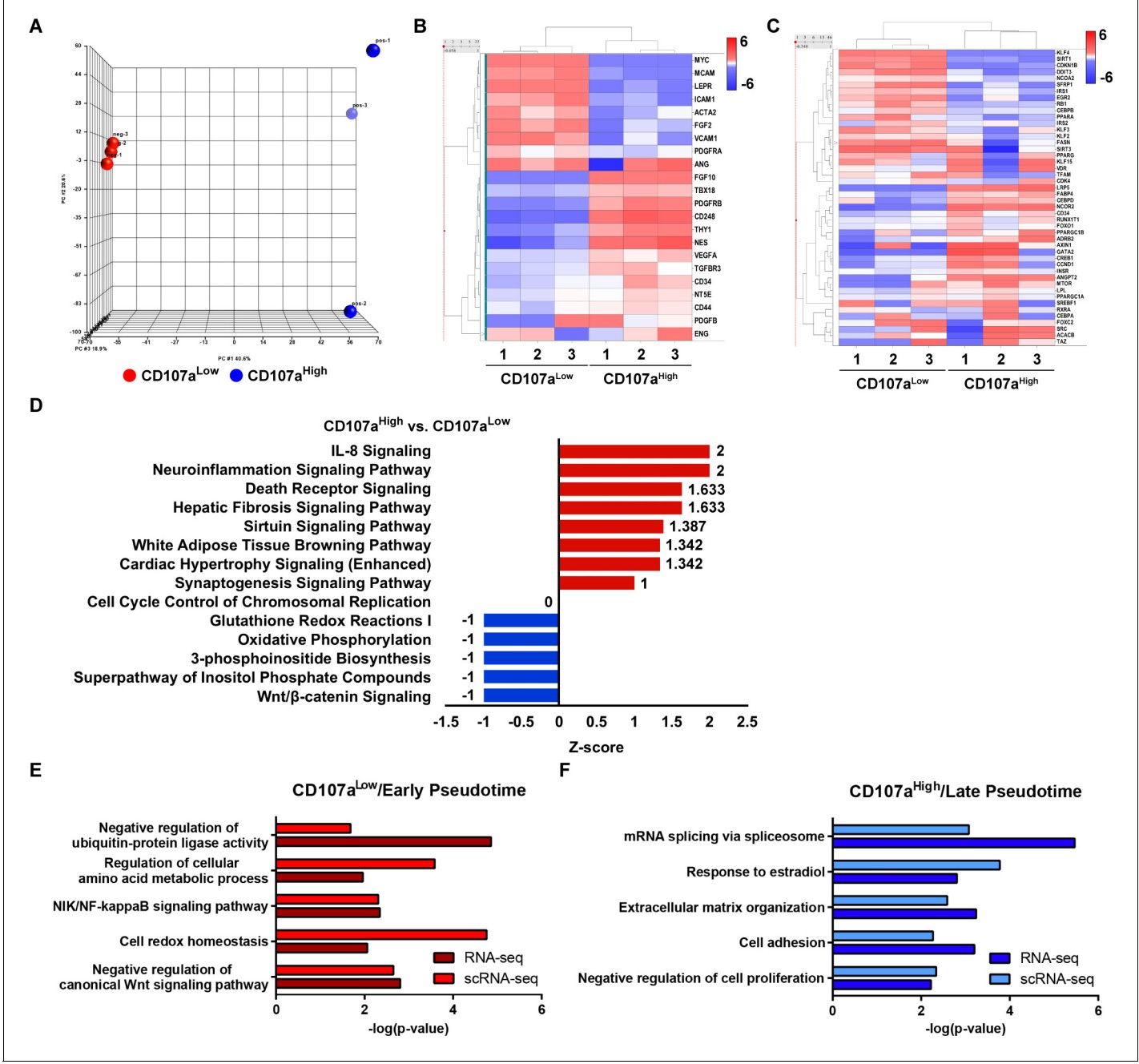

**Figure 4.** Bulk RNA sequencing among uncultured CD107a[low] and CD107a[high] mesenchymal cells and relationship to putative adipose cell hierarchy. (A–F) Total RNA sequencing comparison of CD107a[low]CD31[-]CD45[-] and CD107a[high]CD31[-]CD45[-] mesenchymal cells from a single human subcutaneous WAT sample. (A) Principal component analysis among CD107a[low]CD31[-]CD45[-] and CD107a[high]CD31[-]CD45[-] cells. (B) Heat map demonstrating mRNA expression levels of stemness-related markers and perivascular cell markers among CD107a[low]CD31[-]CD45[-] and CD107a[high]CD31[-]CD45[-] mesenchymal cells. (C) Expression of adipogenic gene markers among CD107a[low]CD31[-]CD45[-] and CD107a[high]CD31[-]CD45[-] mesenchymal cells, shown in heat map. (D) Ingenuity pathway analysis (IPA) identified representative pathways that were upregulated (Z-score >0; red color) or downregulated (Z-score <0; blue color) in CD107a[high]CD31[-]CD45[-] compared with CD107a[low]CD31[-]CD45[-] mesenchymal cells. (E,F) Comparison of CD107a[high/low] bulk sequencing data to human SVF single-cell sequencing data (see again *Figure 3O–Q*). (E) Pathways enriched in both CD107a[low] bulk RNA-seq and early pseudotime genes derived from scRNA-seq. (F) Pathways enriched in both CD107a[high] bulk RNA-seq and late pseudotime genes derived from scRNA-seq.

The online version of this article includes the following figure supplement(s) for figure 4:

**Figure supplement 1.** Additional RNA sequencing analysis of CD107a[low] and CD107a[high] mesenchymal cells.

**Figure supplement 2.** Expression of wnt-related genes among CD107a[low]CD31[-]CD45[-] and CD107a[high]CD31[-]CD45[-] mesenchymal cells.

included terms associated with Wnt signaling as well as energy metabolism. Conversely, GO terms enriched within CD107a$^{high}$ stromal cells were likewise enriched among *GGT5$^+$* cell fractions within 'late' pseudotime (*Figure 4F*). This included terms associated with the regulation of cellular proliferation, cell adhesion, as well as remodeling of extracellular matrix.

## CD107a$^{low}$ rather than CD107a$^{high}$ cells induce ectopic bone formation

We next sought to extend our findings to xenotransplantation studies. If a CD107a$^{low}$ mesenchymal cell population over-represents stem/osteoblast precursor cells, we hypothesized that CD107a$^{low}$ cells would preferentially form ectopic bone within an intramuscular transplantation model (*Figure 5*; *James et al., 2012c*; *Meyers et al., 2018b*). First, CD107a$^{low}$ and CD107a$^{high}$ cell subsets were derived from the same patient sample and mixed mechanically with a demineralized bone matrix

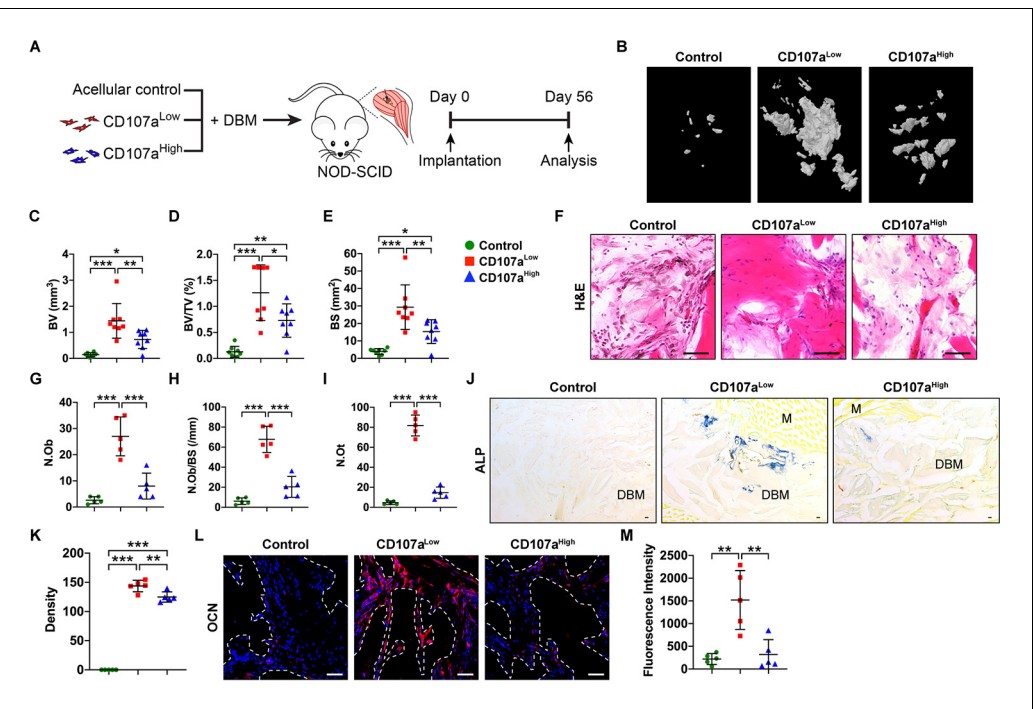

**Figure 5.** CD107a$^{low}$ mesenchymal cells promote ectopic bone formation in vivo. (**A**) FACS-purified CD107a$^{low}$CD31$^-$CD45$^-$ and CD107a$^{high}$CD31$^-$CD45$^-$ mesenchymal cells from the same human subcutaneous WAT sample were implanted intramuscularly in equal numbers in the hindlimbs of NOD-SCID mice. A demineralized bone matrix (DBM) carrier was used, and an acellular control used as a further comparison. Bone formation was assayed after eight wks. Further details on implant composition and animal allocation are found in *Supplementary file 5*. (**B**) Representative micro-computed tomography (μCT) reconstruction images of the implant site among control (DBM only), CD107a$^{low}$, and CD107a$^{high}$ cell grafts. Mineralized bone appears gray. (**C**–**E**) μCT based quantification of ectopic bone formation, including (**C**) Bone volume (BV), (**D**) fractional Bone volume (BV/TV), and (**E**) bone surface (BS). (**F**) Representative histologic appearance by routine H and E of the implant sites among control (DBM only), CD107a$^{low}$, and CD107a$^{high}$ cell grafts. (**G**–**I**) Bone histomorphometric measurements among each treatment group, including (**G**) osteoblast number (N.Ob), (**H**) osteoblast number per bone surface (N.Ob/BS), and (**I**) osteocyte number (N.Ot). (**J**,**K**) Representative alkaline phosphatase (ALP) staining appearing blue (**J**), and photographic quantification (**K**). (**L**,**M**) Representative Osteocalcin (OCN) immunohistochemical staining (**L**), and photographic quantification (**M**). OCN immunostaining appears red, while DAPI nuclear counterstain appears blue. Dots in scatterplots represent values from individual implants, while mean and one SD are indicated by crosshairs and whiskers. M, muscle. *p<0.05; **p<0.01; ***p<0.001. Statistical analysis was performed using a one-way ANOVA followed by Tukey's post hoc test. N = 8 implants per group. Black and white scale bars: 50 μm.

The online version of this article includes the following figure supplement(s) for figure 5:

**Figure supplement 1.** Persistence of human CD107a$^{low}$ and CD107a$^{high}$ cells within intramuscular implants within NOD-SCID mice.

putty carrier (DBX Putty, MTF Biologics) before intramuscular implantation in NOD-SCID mice. The carrier without cells was used as an acellular control. Details of cell implant composition and animal allocation are summarized in *Supplementary file 5*. Intramuscular implants were imaged by micro-computed tomography (µCT) at 8 weeks, demonstrating an accumulation of bone tissue among CD107a[low] laden implants in relation to either CD107a[high] implants or acellular control (*Figure 5A*). Quantitative µCT analysis demonstrated a significant increase in bone volume (BV, 97.7% increase), fractional bone volume (BV/TV, 73.4% increase), and bone surfaces (BS, 91.4% increase) among CD107a[low] as compared to CD107a[high] implants (*Figure 5B–D*). Albeit to a lesser degree, CD107a[high] cells did exhibit bone-forming potential in comparison to acellular control (292.1–473.3% increase in µCT quantitative metrics, *Figure 5B–D*). Histologic analysis revealed conspicuous areas of woven bone among CD107a[low] laden implants, which were not commonly seen among CD107a[high] implants (*Figure 5E*). Bone histomorphometric analysis confirmed these observations, demonstrating significantly increased osteoblast number (N.Ob, 237.5% increase), increased osteoblast number per bone surface (N.Ob/BS, 232.5% increase), and osteocyte number (N.Ot, 460.3% increase) (*Figure 5F–H*). ALP staining and semi-quantitative analysis confirmed an overall increase in serial sections of CD107a[low] treated implants (*Figure 5I,J*, 14.3% increase among CD107a[low] implant sites). Enrichment in the terminal osteogenic differentiation marker osteocalcin (OCN) was also confirmed among CD107a[low] implants, shown by immunostaining and semi-quantitative analysis (*Figure 5K,L*, 345.3% increase among CD107a[low] implant sites). Detection of human nuclear antigen (HNA) among implant sites confirmed the persistence of human cells across both groups, which were overall similar in frequency (*Figure 5—figure supplement 1*).

## CD107a[low] rather than CD107a[high] mesenchymal cells induce spine fusion

Having observed that CD107a[low] cell preparations demonstrate enhanced ectopic bone formation, we next challenged these cells to a posterolateral lumbar spine fusion model within athymic rats (*Figure 6*; *Chung et al., 2014*; *Lee et al., 2015*). CD107a[low] and CD107a[high] cell subsets from patient-identical samples were implanted bilaterally in an L4-L5 spine fusion model (*Figure 6—figure supplement 1*). Details of cell implant composition and animal allocation are summarized in *Supplementary file 6*. A qualitative increase in radiodensity was observed among CD107a[low] treated animals within the spine fusion bed over the post-operative period by high-resolution roentgenography (*Figure 6—figure supplement 2*). Progressive increase in density of the implant sites was confirmed by dual-energy X-ray absorptiometry (DXA)-based quantification, with a gradual and significant increase in CD107a[low] treated spinal implants in comparison to either CD107a[high] treated cells or control (*Figure 6A*). Fusion rate was next assessed by a validated manual palpation scoring (*Figure 6B*; *Grauer et al., 2001*). Consistent with prior studies (*Chung et al., 2014*), after 8 weeks acellular control-treated animals showed 14.3% fusion (1/7 animals). Analyses performed after 8 weeks demonstrated 62.5% spine fusion among CD107a[low] treated animals (6/8 animals). In comparison, CD107a[high] treated animals showed 37.5% fusion (3/8 animals). µCT imaging and reconstructions demonstrated lack of bone bridging within the spinal fusion segments of control-treated and CD107a[high] treated implant sites (*Figure 6C*). In comparison, more robust evidence of bone bridging was observed among CD107a[low] spine fusion segments (*Figure 6C*). Quantitative µCT analysis demonstrated a significant increase in bone volume (BV), fractional bone volume (BV/TV), and bone surfaces (BS) among CD107a[low] implant sites in comparison to acellular control (*Figure 6D–F*, 58.6–80.7% increase across µCT metrics). In contrast, CD107a[high] spine fusion segments demonstrate no statistically significant change in µCT assessments in comparison to acellular control (*Figure 6D–F*, 15.3–25.2% change in comparison to acellular control). These findings were confirmed using histologic and histomorphometric assessments of the spinal fusion segment across treatment groups (*Figure 6G–J*). Histologic analysis revealed conspicuous areas of woven bone among CD107a[low] implants, which were not commonly seen among CD107a[high] implants (*Figure 6G*). Bone histomorphometric analysis confirmed these observations, demonstrating significantly increased osteoblast number (172.2% increase among CD107a[low] implant sites in comparison to CD107a[high] implant sites), increased osteoblast number per bone surface (183.5% increase), and osteocyte number (357.1% increase) (*Figure 6H–J*). ALP enzymatic staining and OCN immunohistochemical staining confirmed the above findings (*Figure 6K–N*). In summary, CD107a[low] but not CD107a[high] mesenchymal cell demonstrate improvements in bone-forming potential across two orthopaedic models.

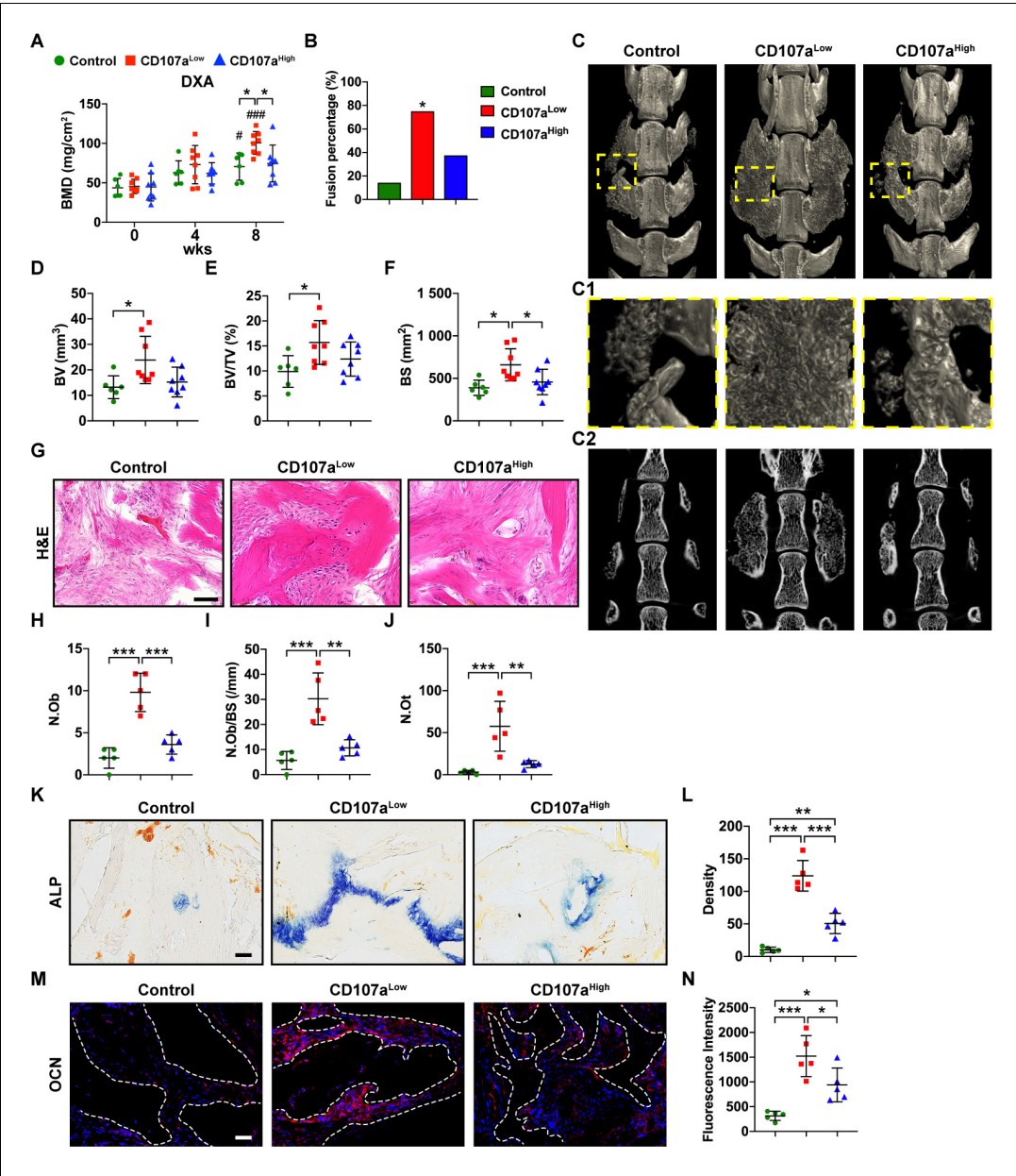

**Figure 6.** CD107a[low] mesenchymal cells induce spine fusion in vivo. FACS-purified CD107a[low]CD31[-]CD45[-] and CD107a[high]CD31[-]CD45[-] mesenchymal cells from the same human subcutaneous WAT sample were implanted in equal numbers in a posterolateral spinal fusion model in athymic rats. A demineralized bone matrix (DBM) carrier was used, and an acellular control used as a further comparison. Animals were analyzed at up to eight wks post-operatively. (**A**) Bone mineral density (BMD) assessed by DXA (dual-energy X-ray absorptiometry) within the lumbar implantation site, at 0, 4, and 8 wks. (**B**) Spine fusion rate, assessed by manual palpation after eight wks. *: CD107a[low] compared with acellular control. (**C**) Representative micro-computed tomography (µCT) reconstruction images of the spine fusion site among CD107a[low] and CD107a[high] treated samples, in comparison to acellular control. Images are shown from the dorsal aspect. (**C1**) Corresponding high magnification µCT reconstruction of the fusion site. (**C2**) Corresponding coronal µCT cross-sectional image. (**D–F**) µCT-based quantification of bone formation within the spine fusion site, including (**D**) Bone volume (BV), (**E**) fractional Bone volume (BV/TV), and (**F**) bone surface (BS). (**G**) Representative histologic appearance by routine H and E of the implant sites among control (DBM only), CD107a[low], and CD107a[high] cell grafts within the spine fusion site. (**H–J**) Bone histomorphometric measurements among each treatment group, including (**H**) osteoblast number (N.Ob), (**I**) osteoblast number per bone surface (N.Ob/BS), and (**J**) osteocyte number (N.Ot). (**K,L**) Representative alkaline phosphatase (ALP) staining appearing blue (**K**), and photographic quantification within the spine fusion site (**L**). (**M**) Representative Osteocalcin

*Figure 6 continued on next page*

*Figure 6 continued*

(OCN) immunohistochemical staining (**M**), and photographic quantification within the spine fusion site (**N**). OCN immunostaining appears red, while DAPI nuclear counterstain appears blue. Dots in scatterplots represent values from individual animal measurements, while mean and one SD are indicated by crosshairs and whiskers. #p<0.05 and ###p<0.001 in relation to corresponding 0 wk timepoint; *p<0.05; **p<0.01; ***p<0.001. Statistical analysis was performed using a two-way ANOVA (**A**) or one-way ANOVA followed by Tukey's post hoc test (**D–N**). N = 6–8 animals per group. Black and white scale bars: 50 μm.

The online version of this article includes the following figure supplement(s) for figure 6:

**Figure supplement 1.** Illustration of procedure for posterolateral lumbar spine fusion in athymic rats.

**Figure supplement 2.** High-resolution roentgenography (XR) demonstrates ossification of spinal fusion implants within CD107a^low treated sites.

## Discussion

Mesenchymal progenitor cells are broadly distributed in post-natal organs, where they are concentrated principally in perivascular areas. Microvascular pericytes were first recognized to include such progenitors since they grow into mesenchymal stem cells in culture (*Crisan et al., 2008*). The outermost layer enwrapping arteries and veins, or *tunica adventitia*, that used to be considered as a mere fibroblast-populated collagen sheath anchoring vessels within tissues, is also home to presumptive MSCs (*Corselli et al., 2012*; *Kramann et al., 2016*). Here, we have used an antibody array targeting all human CD surface markers to identify several novel antigens expressed by human adipose tissue-resident perivascular cells. We found, among other surface antigens, that CD107a, aka LAMP-1, is expressed at the surface of subsets of adventitial cells and pericytes, which was confirmed in terms of gene expression on RNA sequencing libraries, and corroborated by immunohistochemistry, surface CD107a being co-expressed with canonical markers of pericytes and adventitial cells. Further, flow cytometry analysis of the total stromal vascular fraction extracted from adipose tissue showed a continuum of non-endothelial, non-hematopoietic CD107a expressing cells that could be gated back to pericytes and adventitial cells. Altogether, these results confirmed unequivocally that surface CD107a/LAMP-1, used generally as a marker of NK cell activity (*Bryceson et al., 2005*), is present on subsets of human MSC-related perivascular cells, and established the conditions for the purification of these cells by flow cytometry.

Albeit not described before at the surface of human pericytes and other perivascular cells, CD107a has been earlier detected on human bone marrow and dental conventionally cultured MSCs, where it binds the enamel matrix protein amelogenin and in turn induces cell proliferation (*Huang et al., 2010*). Besides, CD146^+CD107a^+ human bone marrow MSCs have been recently described as endowed with the highest immunomodulatory and secretory, hence therapeutic, potential in experimental joint inflammation (*Bowles et al., 2019*). Besides suggested association of surface CD107a with progenitor cell proliferation and immunomodulation related tissue repair (*Bowles et al., 2019*; *Huang et al., 2010*), lysosomal CD107a has also been linked to neural stem cell potential (*Yagi et al., 2010*). Here, we show that surface expression of CD107a divides adipocyte- from osteoblast precursors within human perivasculature. Adipocyte progenitor distribution was virtually confined to the CD107a^high cell subset; in agreement, adipogenic potential in culture was restricted to this cell compartment. However, knockdown of CD107a in ASCs slightly promoted adipogenic differentiation, suggesting that CD107a can be used for identification of functionally relevant subsets, which is likely not explained by intrinsic function of CD107a protein. CD107a is thought to be responsible for maintaining the structural integrity of the lysosomal compartment (*Eskelinen, 2006*). Lysosomes provide the degradative enzymes for autophagy and are involved in autophagy regulation, primarily through its relationship with the master kinase complex, mTORC1 (*Mrschtik and Ryan, 2015*; *Yim and Mizushima, 2020*). Therefore, CD107a may also play an essential role in autophagy, which itself has been shown crucial for adipocyte differentiation (*Guo et al., 2013*). Whether CD107a^high cells enhance adipogenesis by promoting autophagy is an interesting and worthy follow-up topic. Conversely, osteogenic ability in vitro was almost totally restricted to purified CD107a^low cells, which also exhibited the highest CFU-F potential. We confirmed the exclusive bone-forming potential of CD107a^low cells in vivo, in situations of ectopic intramuscular ossification and lumbar spine fusion. Thus, CD107a^low cells likely represent a more progenitor cell, while

CD107a[high] cells are a more mature, differentiated cell population. Similar results are found in the hematopoietic system, where the most primitive hematopoietic stem cells are Thy-1[low], whereas Thy-1[high] cells belong to a well-defined blood cell lineage (*Spangrude et al., 1988*).

The natural function of these ubiquitous perivascular mesenchymal progenitor cells remains a dominant, yet unanswered question. While subsets of mouse pericytes are known, in contexts of experimental disease or injury, to give rise in situ to white adipocytes, follicular dendritic cells, myoblasts, and myofibroblasts (*Dellavalle et al., 2011*; *Dulauroy et al., 2012*; *Krautler et al., 2012*; *Murray et al., 2017*; *Tang et al., 2008*), the osteogenic and chondrogenic potentials present in these microvascular cells are unlikely to be ever used in the turnover and repair of soft tissues. On the other hand, a contribution to blood vessel pathologic remodeling of perivascular presumptive MSCs has been recognized, as osteoblast and smooth muscle cell forerunners for calcific arteriosclerosis and atherosclerosis, respectively (*Kramann et al., 2016*). How such pathogenic side effects exerted by perivascular progenitors are counterbalanced by beneficial contributions to tissue homeostasis is not known, but might be related to differential recruitment of uni- or multipotent progenitors, as well as activation in these cells of mechanisms independent of cell differentiation such as growth factor production or cell contact-mediated immunomodulation (*Pittenger et al., 2019*). A precise phenotypic and functional characterization of perivascular progenitor cells is a requisite for the understanding of the role of these cells in developmental, regenerative, and pathophysiologic processes. In particular, unwrapping the intrinsic heterogeneity of these cells is a prioritized, ongoing process.

Although abundant evidence of the osteogenic potential of perivascular cells exists in humans and mice (*Askarinam et al., 2013*; *Chung et al., 2014*; *Corselli et al., 2012*; *Crisan et al., 2008*; *James et al., 2017*; *James and Péault, 2019*; *James et al., 2012a*; *James et al., 2012a*; *James et al., 2012b*; *James et al., 2012c*; *Kramann et al., 2016*; *Lee et al., 2015*; *Meyers et al., 2018b*; *Meyers et al., 2018b*; *Tawonsawatruk et al., 2016*; *Wang et al., 2020*; *Xu et al., 2019*), there is recent proof that this competence is restricted to discrete cell subsets. For instance, CD10 expression marks a subset of human adipose tissue adventitial cells with higher bone-forming potential (*Ding et al., 2019*), and the highest calcification potential is attributed to mouse adventitial cells co-expressing CD34 and PDGFRα (*Wang et al., 2020*). On the other hand, human perivascular cells expressing the ROR2 Wnt receptor exhibit stronger chondrogenic ability than ROR2 negative counterparts (*Dickinson et al., 2017*). Therefore, the concept is emerging of a developmental micro-heterogeneity of perivascular cells, the surrounding niches being suggested to host a whole hierarchy of mesenchymal progenitor cells already documented to include adipocyte-, chondrocyte-, and osteoblast progenitors. Although the data herein were obtained on abdominal subcutaneous adipose tissue, our preliminary results indicate that the same cell partition can be achieved in other fat depots. Moreover, we have observed the same restriction of adipogenic and osteogenic potentials, respectively, to CD107a[high] and CD107a[low] perivascular cells sorted from the human placenta, which suggests the broad and possibly ubiquitous distribution of these functionally distinct subpopulations. Strikingly, these tissues are not natural sites of ossification, which raises the recurring question of the physiologic significance of these unrelated developmental potentials. It is conceivable that mesodermal cell turnover and regeneration in adult tissues be mediated exclusively by multipotent, MSC like cells, irrelevant potentials in a given tissue being always repressed. It is more difficult to justify that progenitors committed to a given cell compartment be maintained in a tissue devoid of this very cell lineage, such as osteogenic dedicated progenitors in adipose tissue, at the expense of adipogenic cells. Since the hypothesis that these unrelated progenitors can be mobilized through blood circulation to drive regeneration in other organs is not supported for the moment, we can only speculate that such atypical differentiation potentials are irreversibly associated with other tissue repair mechanisms of broader applicability, which remain to be identified. Further characterization of the role of CD107a at the cell surface may contribute to clarifying this issue.

Our in vitro studies, coupled with single-cell transcriptomics, suggest that CD107a, an endolysosome transmembrane protein, traffics to the cell surface during early adipogenesis, suggesting a specific function in this cell lineage. Future studies will tell whether CD107a/LAMP-1, a unique novel marker of the perivascular mesenchymal stem cell hierarchy, will also shed light on the tissue regeneration mechanisms initiated in this niche.

## Materials and methods

### Immunohistochemistry and microscopy

All human tissues were obtained under Johns Hopkins University institutional IRB approval with a waiver of informed consent. For histology, human subcutaneous fat tissue was obtained from three anonymized female donors from the abdominal or thigh area. Human fat tissue was embedded in optimal cutting temperature compound (OCT) (Sakura, Torrance, CA), and cryo-sectioned at 30 µm thickness. For immunofluorescent staining, all sections were blocked with 5% goat serum in PBS for 1 hr at room temperature (RT). The following primary antibodies were used: anti-αSMA (RRID:AB_1951138, 1:100), anti-CD107a (RRID:AB_1727417/RRID:AB_10719137/RRID:AB_470708/RRID:AB_449893, 1:100), anti-CD31 (RRID:AB_448167/RRID:AB_726362, 1:100), anti-CD34 (RRID:AB_1640331, 1:100), anti-CD146 (RRID:AB_2143375, 1:100), or anti-Gli1 (RRID:AB_880198, 1:100; see antibody details in *Supplementary file 7*) for overnight incubation at 4° C. Next, anti-rabbit Alexa Fluor 647-conjugated (RRID:AB_2722623, 1:200), anti-mouse Alexa Fluor 488-conjugated (RRID:AB_2688012, 1:200), anti-goat Alexa Fluor 647-conjugated (RRID:AB_2687955, 1:200), or anti-rat Alexa Fluor 647-conjugated secondary antibodies (RRID:AB_2864291, 1:200, Abcam, San Francisco, CA) were used (incubation 2 hr at RT). DAPI mounting medium was used (RRID:AB_2336788, Vector laboratories, Burlingame, CA), and visualized using a Zeiss 800 confocal microscope (Zeiss, Thornwood, NY). For colorimetric immunohistochemistry staining, sections were blocked with 2.5% horse serum for 20 min at RT. Anti-CD107a primary antibody (RRID:AB_470708, Abcam, 1:100) was added and incubated overnight at 4° C. Next, incubation with alkaline phosphatase (AP) polymer anti-mouse IgG reagent was performed for 30 min (RRID:AB_2336535, MP-5402, Vector laboratories), followed by AP substrate solution (RRID:AB_2336847, SK-5100, Vector laboratories), followed by hematoxylin counterstain and microscopic imaging using a Leica DM6 B microscope (Leica Microsystems Inc, Wetzlar, Germany).

### Adipose-derived stromal cells (ASCs) isolation and FACS isolation of human AT cell populations

For cell isolation, human lipoaspirate was obtained from healthy adult donors and was stored no more than 72 hr at 4°C before processing with some modifications from prior protocols (*Wang et al., 2019*; *Xu et al., 2019*). Patient gender and donor area are shown in *Supplementary file 1*. Equal volume of phosphate-buffered saline (PBS) was used to wash lipoaspirates. The washed lipoaspirate was digested with 1 mg/ml type II collagenase (Washington Biochemical; Lakewood, NJ) in Dulbecco's Modified Eagle's Medium (DMEM) containing 0.5% bovine serum albumin (Sigma-Aldrich, St. Louis, MO) at 37°C for 60 min under agitation, followed by centrifugation to remove adipocytes. The cell pellet was resuspended and incubated in red cell lysis buffer (155 mM $NH_4Cl$, 10 mM KHCO3, and 0.1 mM EDTA) at RT for 10 min. After centrifugation, the stromal vascular fraction (SVF) was resuspended in PBS and filtered at 40 µm. In select studies, SVF was culture- expanded as adipose-derived stromal cells (ASCs) for further evaluation. Human ASCs were cultured and used for experiments at passage 2–6. Fluorescence activated cell sorting (FACS) was next performed using a Beckman MoFlo (Beckman, Indianapolis, IN), with analysis performed using the FlowJo software (RRID:SCR_008520, BD Biosciences, San Jose, CA). Cells were incubated with anti-CD45-allophycocyanin-cyanin 7 (RRID:AB_396891, 1:30; BD Pharmingen, San Diego, CA), anti-CD31-allophycocyanin-cyanin 7 (RRID:AB_2738350, 1:100, BD Pharmingen), anti-CD34 (RRID:AB_11154586, 1:60, BD Pharmingen), anti-CD146 (RRID:AB_324069, 1:100, Bio-Rad, Hercules, CA), and/ or anti-CD107a-allophycocyanin (RRID:AB_1727417, 1:20; BD Pharmingen), for 20 min on ice. Propidium iodide (PI) staining solution (BD Pharmingen) was used to gate out non-viable cells. See *Supplementary file 7* for a list of antibodies used. Gating was performed to isolate either $CD146^+$ pericytes ($CD146^+CD34^-CD31^-CD45^-$), $CD34^+$ adventicytes ($CD34^+CD146^-CD31^-CD45^-$), $CD107a^{low}$ cells ($CD107a^{low}CD31^-CD45^-$) or $CD107a^{high}$ cells ($CD107a^{high}CD31^-CD45^-$). Cell surface markers were analyzed using either Lyoplate (BD Biosciences) or flow cytometry. For flow cytometry, cells were incubated with the following antibodies for 20 min on ice: anti-CD34 PE-CF594 (RRID:AB_11154586), anti-CD146 FITC (RRID:AB_324069), anti-CD44 Alexa Fluor 700 (RRID:AB_10645788), anti-CD73 PE (RRID:AB_2033967), anti-CD90 FITC (RRID:AB_395969), anti-CD105 PE-CF594 (RRID:AB_11154054), and anti-CD107a APC (RRID:AB_1727417, *Supplementary file 7*). For select studies

using culture-expanded cells, flow cytometry was performed after trypsinization and cell re-suspension in HBSS (Life Technologies, Gaithersburg, MD) with 0.5% bovine serum albumin (Sigma-Aldrich). All cells were cultured at 37°C in a humidified atmosphere containing 95% air and 5% $CO_2$. Unless otherwise stated, cells were cultured in Endothelial Cell Growth Medium-2 (EGM-2; Lonza, Gaithersburg, MD).

## Human bone marrow mesenchymal stem cell isolation

Bone marrow mesenchymal stem cells (BMSCs) from anonymized human femur and tibia were isolated using previously reported methods (*Xu et al., 2019*). Marrow cells were flushed with PBS and passed through a 70 µm cell strainer (BD Biosciences) to obtain single cells, which were seeded into T75 flasks. Non-adherent cells were removed after 5 d and medium was changed every 3 d. BMSCs were cultured in growth medium consisting of DMEM, 15% fetal bovine serum (FBS; Gibco, Grand Island, NY), 1% penicillin/streptomycin (Gibco).

## Identification of novel human perivascular cell markers using lyoplate

The BD Lyoplate Human Cell Surface Marker Screening Panel contains 242 purified and lyophilized monoclonal antibodies to cell surface markers, along with AlexaFluor 647 conjugated goat anti-mouse Ig and goat anti-rat Ig secondary antibodies, distributed in three 96-well plates, as well as mouse and rat isotype controls for assessing isotype-specific background. The Lyoplate array was used according to manufacturer's instructions. Aspirated human subcutaneous fat was digested with collagenase, washed by centrifugation and the SVF recovered as described above. After washing and red cell lysis, SVF cells were stained with the following reagents: DAPI 1/100 (1 µg/ml final), FITC-CD146 1/40, PE-CD45 1/20, PE-Cy7-CD34 1/33. Using a multi-channel pipette, 100 µl aliquots of antibody stained SVF (500,000 to 1 million cells) were distributed in the wells, and 50 µl of reconstituted Lyoplate antibody solution were added to each well according to the template. Plates were incubated on ice in the dark for 30 min, then cells were washed twice by adding 100 µl of staining solution to each well and spinning at 300xg for 5 min. 100 µl of 4% paraformaldehyde (PFA) were added to each well and incubated at RT for 30 min. Labeled cells were washed again and either stored at 4°C or analyzed directly on a LSR II flow cytometer (BD Biosciences).

## Transcriptomics

In select experiments, global gene expression analysis of $CD107a^{low}CD31^-CD45^-$ and $CD107a^{high}CD31^-CD45^-$ cells from adipose tissue was performed. The RNA content of $CD107a^{low}$ and $CD107a^{high}$ cells was detected by total RNA sequencing. Briefly, total RNA was extracted from $CD107a^{low}$ and $CD107a^{high}$ cells with Trizol (Life Technologies). After purification and reverse transcription, cDNA samples were sent to the JHMI Transcriptomics and Deep Sequencing Core and quantified by deep sequencing with the Illumina NextSeq 500 platform (Illumina, San Diego, CA). Data were analyzed using software packages including Partek Genomics Suite (RRID:SCR_011860), Spotfire DecisionSite with Functional Genomics (RRID:SCR_008858), and QIAGEN Ingenuity Pathway Analysis (RRID:SCR_008653).

## Single-cell RNA sequencing (scRNA-seq)

ScRNA-seq data were obtained from the Gene Expression Omnibus (GEO) repository, accession number GSE128889 (GSM3717979, GSM3717977). Initial quality control removed cells expressing >200 and<6000 genes and a mitochondrial content >5%. Data normalization, dimensional reduction and clustering were conducted in Seurat (RRID:SCR_016341) as previously described in the original publication with the exception of altered clustering resolutions. Trajectory plots were generated in Monocle (RRID:SCR_018685) as previously described. For exocytosis pathway activation, a gene list of exocytosis activating/promoting genes was generated from the previously annotated KEGG pathway. Gene lists were filtered for genes that met the Monocle cutoff criteria as an expressed gene (expressed in a minimum of 10 cells). Expression levels were normalized first for individual cell UMI counts, then to the average gene expression across the whole sample population. This normalized expression across pseudotime was displayed following average with nearest neighbors. Values above one indicate enriched expression of genes associated with exocytosis while values below one indicate expression below population averages. Pathway analyses were conducted

on CD107a$^{low}$ and CD107a$^{high}$ cells from bulk RNA-seq experiments, as well as on early- and late-expressing, pseudotemporally-regulated genes derived from scRNA-seq data.

## Osteogenic differentiation assay and ALP/Alizarin red staining

For osteogenic differentiation, cells were seeded at the density of $2.5 \times 10^5$/ml in 12- or 24-well plates. Upon confluency, medium was changed to osteogenic differentiation medium, composed of DMEM, 10% FBS (Gibco), 1% penicillin/streptomycin, with 50 µM ascorbic acid, 10 mM β-glycero-phosphate, and 100 nM dexamethasone (Sigma-Aldrich) (*Xu et al., 2019*). Medium was changed every 3 d. Leukocyte Alkaline Phosphatase Kit (Sigma-Aldrich) was used for alkaline phosphatase staining at 3 or 7 d of differentiation. Image J (RRID:SCR_003070) was used to detect the integrated density of ALP staining. Alizarin red S (Sigma-Aldrich) was used to stain cultures at 7 or 10 d of differentiation to detect mineralization. Next, calcium precipitate was dissolved with 0.1N sodium hydroxide and quantified by absorbance at 548 nm. Experiments were performed in N = 3 biological replicates, and in experimental triplicates in each case.

## Adipogenic differentiation assay and oil red O staining

For adipogenic differentiation, cells were seeded at the density of $2.5 \times 10^5$/ml in 12- or 24-well plates. Upon subconfluency, adipogenic differentiation medium (DMEM, 1% penicillin/streptomycin, 10% FBS with 1 µM dexamethasone, 200 µM indomethacin, 10 µg/ml insulin, and 500 µM 3-isobutyl-1-methylxanthine (Sigma-Aldrich)) was used. Medium was changed every three d. Cells were fixed with 4% PFA and stained with Oil Red O solution at 5 to 7 d of differentiation (*Meyers et al., 2018a*). Experiments were performed in N = 3 biological replicates, and in experimental triplicates in each case.

## Chondrogenic differentiation assay and Alcian blue staining

For chondrogenic differentiation, cells were seeded at high-density micromass environment ($1 \times 10^7$/ml, 10 µl/drop) in 12-well plates and cultured in 37°C. After 4 hr, chondrogenic differentiation medium (DMEM, 1% penicillin/streptomycin, 10% FBS with 10 ng/ml transforming growth factor-β3 (R and D Systems, Minneapolis, MN), 100x ITS+ Premix (Corning Incorporated, Corning, NY), 50 µg/ml ascorbic acid, 40 µg/ml proline, 100 µg/ml pyruvate, and 100 nM dexamethasone (Sigma-Aldrich)) was added. Medium was changed every three d. Cells were fixed with 4% PFA and embedded in OCT for cryosectioning at 18 µm thickness. Slides were stained with Alcian Blue and Fast Red.

## Proliferation assay

Proliferation assays were performed in 96 well plates ($2 \times 10^3$ cells/well) and measured at 72 hr using the CellTiter96 AQueous One Solution Cell Proliferation Assay kit (MTS, G358A; Promega, Madison, WI). Briefly, 20 µl of MTS solution was added to each well and incubated for 1 hr at 37°C. The absorbance was assayed at 490 nm using Epoch microspectrophotometer (Bio-Tek, Winooski, VT).

## CFU assay

For all CFU assays, 1,000 cells / well were seeded in 6-well plates. For CFU-F analysis, cells were cultivated for 14 d in growth medium, fixed with 100% methanol and stained with 0.5% crystal violet. For CFU-OB assays, cells were cultivated for 7 d in growth medium, followed by 3 d culture in osteogenic differentiation medium followed by alkaline phosphatase staining. For CFU-AD assays, cells were cultured for 8 d in growth medium, followed by 8 d in adipogenic differentiation medium, followed by Oil red O staining. For quantification, the total number of positive colonies was calculated per well. All CFU assays were performed with N = 6 wells per group.

## Immunocytochemistry

Cells were seeded at the density of $1.5 \times 10^5$/ml in EZ SLIDE (Merck Millipore, Billerica, MA). After confluence, medium was changed to osteogenic differentiation medium or adipogenic differentiation medium for 3 d. To visualize the membranous expression of CD107a, cells were directly stained with anti-CD107a-allophycocyanin (RRID:AB_1727417, 1:20; BD Pharmingen) for 20 min at 4°C. Then

cells were washed with PBS and fixed with 4% PFA, followed by DAPI mounting medium (RRID:AB_2336788, Vector laboratories).

## Vacuolin-1 treatment

To inhibit exocytosis, cells were pre-treated with vacuolin-1 (1 µM; Sigma-Aldrich) for 24 hr before adipogenic differentiation medium was added. Cells were incubated in the presence or absence of vacuolin-1 (1 µM) in adipogenic differentiation medium for 3 d. Surface CD107a expression was detected by flow cytometry or confocal microscopy (Zeiss 800). For immunofluorescence staining, the cell membrane was labeled using Wheat Germ Agglutinin Conjugates (Thermo Fisher Scientific, Waltham, MA).

## siRNA knockdown

In select experiments, siRNA-mediated knockdown of *LAMP1* was performed among primary human ASCs prior to osteogenic or adipogenic differentiation. *LAMP1* siRNA (Cat# s8080 and s8082) and negative control siRNA (Cat# 4390843) were obtained from Thermo Fisher Scientific. TransIT-LT1 Transfection Reagent (Mirus Bio, Madison, WI) was used as described by the manufacturer. The medium was changed after 4 hr. Validation by qRT-PCR was performed.

## Real-time polymerase chain reaction

Gene expression was analyzed with quantitative real-time polymerase chain reaction (qRT-PCR) (*Xu et al., 2019*). Total RNA was isolated from cells using TRIzol (Life Technologies). Next, RNA was reverse transcribed into cDNA by iScript cDNA Synthesis Kit (Bio-Rad) following manufacturer's instructions. SYBR Green PCR Master Mix (Life Technologies) was used for RT-PCR. See *Supplementary file 8* for primer information. N = 3 wells per group were used, with all studies performed in biologic triplicates.

## Western blot

Proteins were extracted from cultured cells following lysis in ice cold RIPA buffer (Thermo Scientific) with protease inhibitor cocktail (Cell Signaling Technology, Danvers, MA, USA). Proteins were separated by SDS–polyacrylamide gel electrophoresis and transferred onto a nitrocellulose membrane. The blotted nitrocellulose membranes were blocked with 5% bovine serum albumin for 1 hr and then probed with primary antibodies at 4℃ overnight. Finally, membranes were incubated with a horseradish-peroxidase (HRP)-conjugated secondary antibody and detected by ChemiDoc XRS+ System (Bio-rad). Quantification of protein bands was performed using Image J software (RRID:SCR_003070).

## Intramuscular implantation

Animals were housed and experiments were performed in accordance with institutional guidelines at Johns Hopkins University under ACUC approval. A DBX putty (courtesy of Musculoskeletal Transplant Foundation, Edison, NJ) was used for ectopic bone formation in mice. Briefly, CD107a[low] or CD107a[high] cells derived from the same human WAT sample at passage five were prepared at a density of 3 million total cells in 40 µl PBS and mechanically mixed with 45 mg DBX putty. DBX alone was used as an acellular control. The cell preparation was then implanted intramuscularly into the thigh muscle pouch of 8-week-old male NOD-SCID mice (RRID:IMSR_JAX:001303, The Jackson Laboratory, Bar Harbor, ME) as previously described with some modifications (*James et al., 2012c*). Mice were anesthetized by isoflurane inhalation and premedicated with buprenorphine. Incisions in the hindlimbs were made, and pockets were cut in the biceps femoris muscles by blunt dissection, parallel to the muscle fiber long axis. Dissection methods and the surgical manipulation of tissues were kept as constant as possible across animals. The muscle and skin were each closed with 4–0 Vicryl*Plus sutures (Ethicon Endo-Surgery, Blue Ash, OH). See *Supplementary file 5* for an outline of animals per experimental group. Surgical implantations and subsequent analyses were performed blinded to treatment group.

## Lumbar spine fusion

Posterolateral lumbar spinal fusion was performed on 23-week-old male athymic rats (RRID:RGD_2312499) as previously described (*Chung et al., 2014*). Posterior midline incisions were made over the caudal portion of the lumbar spine, and two separate fascial incisions were made 4 mm bilaterally from the midline. L4 and L5 lumbar spines were exposed by blunt muscle splitting technique and decorticated using a low speed burr and micro-drill (Roboz Surgical Instrument Co., Gaithersburg, MD). Next, DBX (300 µl per side) mixed with CD107a$^{low}$ or CD107a$^{high}$ cells from the same AT sample (1.5 million cells, P3-5 passage) or DBX alone were implanted between the transverse processes bilaterally into the paraspinal muscle bed. Finally, the fasciae and skin were each closed using continuous suture (4–0 Vicryl*Plus, Ethicon Endo-Surgery). In vivo imaging was performed by a combination X-ray/DXA (Faxitron Bioptics, Tucson, AZ) at 0, 4, and 8 weeks after surgery. Rats were sacrificed 8 weeks after surgery, and the spines were harvested for further analysis. See *Supplementary file 6* for animals per experimental group. Surgical procedures and subsequent analyses were performed blinded to treatment group.

## Post mortem analyses

Samples were fixed in 4% PFA for 24–48 hr and evaluated using a high-resolution micro-computed tomography (µCT) imaging system (SkyScan 1275; Bruker MicroCT N.V, Kontich, Belgium). For intramuscular implants, scans were obtained at an image resolution of 15 µm with a 1 mm of aluminum filter (X-ray voltage of 65 kVP, anode current of 153 uA, exposure time of 218 ms). For spine fusion samples, scans were obtained at an image resolution of 22 µm with a 1 mm of aluminum filter (X-ray voltage of 55 kVP, anode current of 181 uA, exposure time of 218 ms). NRecon software (SkyScan, Bruker) was used to reconstruct images from the 2D X-ray projections. For the 3D morphometric analyses of images, CTVox and CTAn were used (SkyScan, Bruker). For muscle pouch implantation and spine fusion analysis, volumes of interest were shaped to encompass all the implant and exclude native bone, with a threshold value of 65.

After radiographic imaging, samples were transferred to 14% EDTA for decalcification for 28–60 d. Samples were then embedded in OCT for cryosectioning at 18 µm thickness. H and E and ALP staining were performed on serial sections (Leukocyte Alkaline Phosphatase Kit, Sigma-Aldrich). For immunofluorescent staining, sections were washed with PBS three times and blocked with 5% goat serum in PBS for 1 hr at 25° C. Antigen retrieval was performed by trypsin enzymatic antigen retrieval solution (ab970; Abcam) for 10 min at 25° C. Primary anti-osteocalcin antibody (RRID:AB_10675660, 1:100, Abcam) was added to each section and incubated at 4° C overnight. Next, an Alexa Fluor 647 goat anti-rabbit IgG (H+L) polyclonal (RRID:AB_2722623, 1:200, Abcam) was used as the secondary antibody. Sections were counterstained with DAPI mounting medium (RRID:AB_2336788, Vector laboratories).

## Statistical analysis

Quantitative data are expressed as mean ±one SD. Statistical analyses were performed using the SPSS16.0 software (RRID:SCR_002865) or GraphPad Prism (RRID:SCR_002798, Version 7.0). Our in vitro studies comparing CD107a$^{low}$ to CD107a$^{high}$ cells resulted in effect size of 3.75. Based on $\alpha = 0.05$ and power = 0.8, statistical significance should be observed with N = 6 animals per group assuming a one-way ANOVA with a 0.05 significance level. Student's t-test was used for two-group comparisons, and one-way ANOVA test was used for comparisons of three or more groups, followed by Tukey's post hoc test. Two-way ANOVA test was used for comparisons of two or three groups with different time points. Differences were considered significant with *p<0.05, **p<0.01, and ***p<0.001.

## Acknowledgements

We thank the JHU microscopy core facility, JHMI deep sequencing and microarray core facility, and Hao Zhang within the JHU Bloomberg Flow Cytometry and Immunology Core. The slides of human adipose tissue depots used in *Figure 1—figure supplement 2* were donated by Drs. Trivia Frazier and Jeffrey Gimble at Obatala Sciences, Inc (New Orleans LA). AWJ was supported by the NIH/NIAMS (R01 AR070773, K08 AR068316), NIH/NIDCR (R21 DE027922), Department of Defense

(W81XWH-18-1-0121, W81XWH-18-1-0336, W81XWH-18–10613), American Cancer Society (Research Scholar Grant, RSG-18-027-01-CSM), the Maryland Stem Cell Research Foundation, and MTF Biologics. In addition, MTF Biologics donated reagents for the study. This work was also supported by grants from BIRAX (the Britain Israel Research and Academic Exchange Partnership) and the British Heart Foundation (BHF) to BP. The content is solely the responsibility of the authors and does not necessarily represent the official views of the National Institute of Health, Department of Defense, or U.S. Army.

## Additional information

### Competing interests

Bruno Péault: BP is the inventor of perivascular stem cell related patents held by the UC Regents (Patent number: 9730965) and is on the editorial board of Stem Cells. Aaron W James: AWJ is a paid consultant for Novadip. This arrangement has been reviewed and approved by the JHU in accordance with its conflict of interest polices. AWJ receives funding for unrelated research from MTF Biologics and Novadip, and is on the editorial board of American Journal of Pathology and Bone Research. AWJ is the inventor of methods to purify CD107a progenitor cells held by the Johns Hopkins University. The other authors declare that no competing interests exist.

### Funding

| Funder | Grant reference number | Author |
| --- | --- | --- |
| National Institute of Arthritis and Musculoskeletal and Skin Diseases | R01 AR070773 | Aaron W James |
| National Institute of Arthritis and Musculoskeletal and Skin Diseases | K08 AR068316 | Aaron W James |
| National Institute of Dental and Craniofacial Research | R21 DE027922 | Aaron W James |
| Department of Defense | W81XWH-18-1-0121 | Aaron W James |
| American Cancer Society | Research Scholar Grant | Aaron W James |
| American Cancer Society | RSG-18-027-01-CSM | Aaron W James |
| Maryland Stem Cell Research Fund | | Aaron W James |
| MTF Biologics | | Aaron W James |
| British Israel Research Academic Exchange Partnership | | Bruno Péault |
| British Heart Foundation | | Bruno Péault |
| Department of Defense | W81XWH-18-1-0121 | Aaron W James |
| Department of Defense | W81XWH-18-1-0336 | Aaron W James |
| Department of Defense | W81XWH-18-10613 | Aaron W James |

The funders had no role in study design, data collection and interpretation, or the decision to submit the work for publication.

### Author contributions

Jiajia Xu, Conceptualization, Data curation, Validation, Investigation, Methodology, Writing - original draft, Writing - review and editing; Yiyun Wang, Yongxing Gao, Validation, Investigation, Writing - original draft; Ching-Yun Hsu, Software, Investigation, Methodology; Stefano Negri, Data curation, Validation, Investigation, Methodology; Robert J Tower, Software, Validation, Investigation; Ye Tian, Validation, Investigation, Methodology; Takashi Sono, Investigation, Methodology; Carolyn A Meyers, Data curation, Validation, Investigation; Winters R Hardy, Leslie Chang, Shuaishuai Hu,

Nusrat Kahn, Validation, Investigation; Kristen Broderick, Resources; Bruno Péault, Conceptualization, Supervision, Investigation, Visualization, Methodology, Project administration, Writing - review and editing; Aaron W James, Conceptualization, Supervision, Funding acquisition, Validation, Investigation, Methodology, Writing - original draft, Project administration, Writing - review and editing

### Author ORCIDs
Jiajia Xu https://orcid.org/0000-0002-6084-2029
Aaron W James https://orcid.org/0000-0002-2002-622X

### Ethics
Human subjects: All human tissues were obtained under Johns Hopkins University institutional IRB approval with a waiver of informed consent (Approval No. IRB00119905 and IRB00137530).
Animal experimentation: All animal experiments were performed according to the approved protocol of the Animal Care and Use Committee (ACUC) at Johns Hopkins University (Approval No. MO19M266 and RA19M268).

### Decision letter and Author response
Decision letter https://doi.org/10.7554/eLife.58990.sa1
Author response https://doi.org/10.7554/eLife.58990.sa2

## Additional files
### Supplementary files
• Supplementary file 1. Frequency of CD107a$^{low}$CD31$^-$CD45$^-$ or CD107a$^{high}$CD31$^-$CD45$^-$ cells by FACS sorting. Percentages are based on cell frequency within the PI- cell population.

• Supplementary file 2. CD34 frequency among freshly isolated CD107a$^{low}$ and CD107a$^{high}$ cells.

• Supplementary file 3. CD146 frequency among freshly isolated CD107a$^{low}$ and CD107a$^{high}$ cells.

• Supplementary file 4. Canonical mesenchymal stem cell (MSC) markers among freshly isolated CD107a$^{low}$ and CD107a$^{high}$ cells.

• Supplementary file 5. Animal allocation for intramuscular implantation using 12 NOD-SCID mice. Implants were placed bilaterally, with each animal receiving the same treatment on either hindlimb.

• Supplementary file 6. Animal allocation for posterolateral lumbar spine fusion model in athymic rats. Cell-augmented grafts were placed bilaterally on either side of the lumbar spine, with scaffold and cell numbers per side shown.

• Supplementary file 7. Antibodies used.

• Supplementary file 8. Primers used.

• Supplementary file 9. Uncropped versions of representative western blot images from *Figure 3—figure supplement 4B*.

• Transparent reporting form

### Data availability
Expression data that support the findings of this study have been deposited in Gene Expression Omnibus (GEO) with the accession codes GSE148519 and GSE128889 (GSM3717979, GSM3717977).

The following dataset was generated:

| Author(s) | Year | Dataset title | Dataset URL | Database and Identifier |
|---|---|---|---|---|
| James AW, Xu J | 2020 | Expression data of CD107aLow and CD107aHigh cells isolated from human adipose tissue | https://www.ncbi.nlm.nih.gov/geo/query/acc.cgi?acc=GSE148519 | NCBI Gene Expression Omnibus, GSE148519 |

The following previously published dataset was used:

| Author(s) | Year | Dataset title | Dataset URL | Database and Identifier |
|---|---|---|---|---|
| Seale P, Merrick D, Sakers A | 2019 | Identification of a mesenchymal progenitor cell hierarchy in adipose tissue | https://www.ncbi.nlm.nih.gov/geo/query/acc.cgi?acc=GSE128889 | NCBI Gene Expression Omnibus, GSE128889 |

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

# Appendix 1

**Appendix 1—key resources table**

| Reagent type (species) or resource | Designation | Source or reference | Identifiers | Additional information |
|---|---|---|---|---|
| Strain, strain background (mouse) | NOD-SCID | Jackson Laboratory | Strain #001303, RRID:IMSR_JAX:001303 | Male, 8 week old |
| Strain, strain background (rat) | RNU Nude Rat | Charles River Laboratories Inc | Strain #316, RRID:RGD_2312499 | Male, 23 week old |
| Transfected construct (human) | Negative Control siRNA | ThermoFisher Scientific | Cat#4390843 | |
| Transfected construct (human) | LAMP1 siRNA | ThermoFisher Scientific | Cat#4392420 Assay ID s8082 | |
| Transfected construct (human) | LAMP1 siRNA 2# | ThermoFisher Scientific | Cat#4392420 Assay ID s8080 | |
| Antibody | anti-Human CD31-APC-Cy7 (Mouse monoclonal) | BD Pharmingen | Cat# 563653, RRID:AB_2738350 | FACS/Flow cytometry (1:100) |
| Antibody | anti-Human CD31 (Mouse monoclonal) | Abcam | Cat# ab24590, RRID:AB_448167 | Immunofluorescent staining (1:100) |
| Antibody | anti-Human CD31 (Rabbit polyclonal) | Abcam | Cat# ab28364, RRID:AB_726362 | Immunofluorescent staining (1:100) |
| Antibody | anti-Human CD34-PE-CF594 (Mouse monoclonal) | BD Pharmingen | Cat# 562383, RRID:AB_11154586 | Flow cytometry (1:60) |
| Antibody | anti-Human CD34 (Rabbit monoclonal) | Abcam | Cat# ab81289, RRID:AB_1640331 | Immunofluorescent staining (1:100) |
| Antibody | anti-Human CD44-AF700 (Mouse monoclonal) | BD Pharmingen | Cat# 561289, RRID:AB_10645788 | Flow cytometry (1:20) |
| Antibody | anti-Human CD45-APC-Cy7 (Mouse monoclonal) | BD Pharmingen | Cat# 557833, RRID:AB_396891 | FACS/Flow cytometry (1:30) |
| Antibody | anti-Human CD73-PE (Mouse monoclonal) | BD Pharmingen | Cat# 561014, RRID:AB_2033967 | Flow cytometry (1:5) |
| Antibody | anti-Human CD90-FITC (Mouse monoclonal) | BD Pharmingen | Cat# 555595, RRID:AB_395969 | Flow cytometry (1:20) |
| Antibody | anti-Human CD105-PE-CF594 (Mouse monoclonal) | BD Pharmingen | Cat# 562380, RRID:AB_11154054 | Flow cytometry (1:20) |
| Antibody | anti-Human CD107a-APC (Mouse monoclonal) | BD Pharmingen | Cat# 560664, RRID:AB_1727417 | FACS/Flow cytometry (1:20)/Immunocytochemistry (1:100) |

*Continued on next page*

*Appendix 1—key resources table continued*

| Reagent type (species) or resource | Designation | Source or reference | Identifiers | Additional information |
|---|---|---|---|---|
| Antibody | anti-Human CD107a (Mouse monoclonal) | R and D Systems | Cat# MAB4800, RRID:AB_10719137 | Immunofluorescent staining (1:100) |
| Antibody | anti-Human CD107a (Mouse monoclonal) | Abcam | Cat# ab25630, RRID:AB_470708 | Immunohistochemistry (1:100)/ Western blot (1:1000) |
| Antibody | anti-Human CD107a (Rat monoclonal) | Abcam | Cat# ab25245, RRID:AB_449893 | Immunofluorescent staining (1:100) |
| Antibody | anti-Human CD146-FITC (Mouse monoclonal) | Bio-Rad | Cat# MCA2141F, RRID:AB_324069 | Flow cytometry (1:100) |
| Antibody | anti-Human CD146 (Rabbit monoclonal) | Abcam | Cat# ab75769, RRID:AB_2143375 | Immunofluorescent staining (1:100) |
| Antibody | anti-human GAPDH (Rabbit monoclonal) | Cell Signaling Technology | Cat# 5174, RRID:AB_10622025 | Western blot (1:1000) |
| Antibody | anti-human Gli1 (Rabbit polyoclonal) | Abcam | Cat# ab49314, RRID:AB_880198 | Immunofluorescent staining (1:100) |
| Antibody | anti-human Nuclei (Mouse monoclonal) | Sigma-Aldrich | Cat# MAB1281, RRID:AB_94090 | Immunofluorescent staining (1:500) |
| Antibody | anti-human Osteocalcin (Rabbit polyclonal) | Abcam | Cat# ab93876, RRID:AB_10675660 | Immunofluorescent staining (1:100) |
| Antibody | anti-human αSMA (Rabbit polyclonal) | Abcam | Cat# ab21027, RRID:AB_1951138 | Immunofluorescent staining (1:100) |
| Antibody | Anti-rabbit IgG, HRP-linked (Goat polyclonal) | Cell Signaling Technology | Cat# 7074, RRID:AB_2099233 | Western blot (1:5000) |
| Antibody | Anti-mouse IgG, HRP-linked (Horse polyclonal) | Cell Signaling Technology | Cat# 7076, RRID:AB_330924 | Western blot (1:5000) |
| Antibody | anti-mouse IgG H and L-AF488 (Goat polyclonal) | Abcam | Cat# ab150117, RRID:AB_2688012 | Immunofluorescent staining (1:200) |
| Antibody | anti-rabbit IgG H and L-AF488 (Goat polyclonal) | Abcam | Cat# ab150077, RRID:AB_2630356 | Immunofluorescent staining (1:200) |
| Antibody | anti-rabbit IgG H+L-DyLight 594 (Goat) | Vector Laboratories | Cat# DI-1594, RRID:AB_2336413 | Immunofluorescent staining (1:200) |

*Continued on next page*

*Appendix 1—key resources table continued*

| Reagent type (species) or resource | Designation | Source or reference | Identifiers | Additional information |
|---|---|---|---|---|
| Antibody | anti-goat IgG H and L-AF647 (Donkey polyclonal) | Abcam | Cat# ab150135, RRID:AB_2687955 | Immunofluorescent staining (1:200) |
| Antibody | anti-mouse IgG H and L-AF647 (Goat polyclonal) | Abcam | Cat# ab150119, RRID:AB_2811129 | Immunofluorescent staining (1:200) |
| Antibody | anti-rabbit IgG H and L-AF647 (Goat polyclonal) | Abcam | Cat# ab150079, RRID:AB_2722623 | Immunofluorescent staining (1:200) |
| Antibody | anti-rat IgG H and L-AF647 (Goat polyclonal) | Abcam | Cat# ab150167, RRID:AB_2864291 | Immunofluorescent staining (1:200) |
| Antibody | ImmPRESS-AP Anti-Mouse Reagent antibody (Horse) | Vector Laboratories | Cat# MP-5402, RRID:AB_2336535 | Immunohistochemistry (200 µl) |
| Sequence-based reagent | ACAN_F | This paper | PCR primers | AGGCTGGGGAGAGAACTGAAAAG |
| Sequenced-based reagent | ACAN_R | This paper | PCR primers | GCTCACAATGGGGTATCTGACAG |
| Sequenced-based reagent | ACTB_F | This paper | PCR primers | CTGGAACGGTGAAGGTGACA |
| Sequenced-based reagent | ACTB_R | This paper | PCR primers | AAGGGACTTCCTGTAACAATGCA |
| Sequenced-based reagent | ALPL_F | This paper | PCR primers | ACCACCACGAGAGTGAACCA |
| Sequenced-based reagent | ALPL_R | This paper | PCR primers | CGTTGTCTGAGTACCAGTCCC |
| Sequenced-based reagent | COL2A1_F | This paper | PCR primers | CCGCGGTGAGCCATGATTCG |
| Sequenced-based reagent | COL2A1_R | This paper | PCR primers | CAGGCCCAGGAGGTCCTTTGGG |
| Sequenced-based reagent | COMP_F | PMID:23382851 | PCR primers | CAACTGTCCCCAGAAGAGCAA |
| Sequenced-based reagent | COMP_R | PMID:23382851 | PCR primers | TGGTAGCCAAAGATGAAGCCC |
| Sequenced-based reagent | FABP4_F | This paper | PCR primers | ACGAGAGGATGATAAACTGGTGG |
| Sequenced-based reagent | FABP4_R | This paper | PCR primers | GCGAACTTCAGTCCAGGTCAAC |
| Sequenced-based reagent | GAPDH_F | PMID:31482845 | PCR primers | CTGGGCTACACTGAGCACC |
| Sequenced-based reagent | GAPDH_R | PMID:31482845 | PCR primers | AAGTGGTCGTTGAGGGCAATG |
| Sequenced-based reagent | LAMP1_F | This paper | PCR primers | GTCTTCTTCGTGCCGGCGT |
| Sequenced-based reagent | LAMP1_R | This paper | PCR primers | GCAGGTCAAAGGTCATGTTCTT |

*Appendix 1—key resources table continued*

| Reagent type (species) or resource | Designation | Source or reference | Identifiers | Additional information |
|---|---|---|---|---|
| Sequenced-based reagent | LPL_F | This paper | PCR primers | TTGCAGAGAGAGGACTCGGA |
| Sequenced-based reagent | LPL_R | This paper | PCR primers | GGAGTTGCACCTGTATGCCT |
| Sequenced-based reagent | SPP1_F | This paper | PCR primers | CCTCCTAGGCATCACCTGTG |
| Sequenced-based reagent | SPP1_R | This paper | PCR primers | CCACACTATCACCTCGGCC |
| Sequenced-based reagent | RUNX2_F | PMID:31482845 | PCR primers | TGGTTACTGTCATGGCGGGTA |
| Sequenced-based reagent | RUNX2_R | PMID:31482845 | PCR primers | TCTCAGATCGTTGAACCTTGCTA |
| Sequenced-based reagent | PPARG_F | This paper | PCR primers | GACAGGAAAGACAACAGACAAATC |
| Sequenced-based reagent | PPARG_R | This paper | PCR primers | GGGGTGATGTGTTTGAACTTG |
| Sequenced-based reagent | RUNX2_F | This paper | PCR primers | TGGTTACTGTCATGGCGGGTA |
| Sequenced-based reagent | RUNX2_R | This paper | PCR primers | TCTCAGATCGTTGAACCTTGCTA |
| Sequenced-based reagent | SOX9_F | This paper | PCR primers | GAGGAAGTCGGTGAAGAACG |
| Sequenced-based reagent | SOX9_R | This paper | PCR primers | ATCGAAGGTCTCGATGTTGG |
| Commercial assay or kit | Leukocyte Alkaline Phosphatase Kit | Sigma-Aldrich | Cat#85L2-1KT | |
| Chemical compound, drug | Vacuolin-1 | Sigma-Aldrich | Cat#673000 | |
| Chemical compound, drug | TransIT-LT1 Transfection Reagent | Mirus Bio | Cat#MIR2300 | Transfection |
| Software, algorithm | FlowJo | FlowJo | RRID:SCR_008520 | |
| Software, algorithm | ImageJ | NIH | RRID:SCR_003070 | |
| Software, algorithm | Prism | GraphPad | RRID:SCR_002798 | |
| Software, algorithm | Seurat | Seurat | RRID:SCR_016341 | |
| Software, algorithm | Monocle 3 | Monocle | RRID:SCR_018685 | |
| Software, algorithm | Partek Genomics Suite | Partek | RRID:SCR_011860 | |
| Software, algorithm | Spotfire | Spotfire | RRID:SCR_008858 | |
| Software, algorithm | Ingenuity Pathway Analysis | QIAGEN | RRID:SCR_008653 | |
| Software, algorithm | SPSS | SPSS | RRID:SCR_002865 | |

*Continued on next page*

*Appendix 1—key resources table continued*

| Reagent type (species) or resource | Designation | Source or reference | Identifiers | Additional information |
|---|---|---|---|---|
| Other | DAPI stain | Vector Laboratories | Cat# H-1500, RRID:AB_2336788 | |
| Other | Vector Red Substrate Kit | Vector Laboratories | Cat# SK-5100, RRID:AB_2336847 | |

