## [Decision Letter]

**Acceptance summary:**

The reviewers and the editors are in general agreement that this is a well-designed and executed study that evaluated the usefulness of CD170a as a surrogate marker to distinguish between osteoblast precursors and adipocytes within the human perivascular tissue. The authors have clearly demonstrated the osteogenic potential of CD107a^low^ expressing cells, both in vitro and in vivo. Furthermore, they have carefully and thoroughly addressed the reviewer suggestions and made appropriate amendments in the revised paper. The precise phenotypic and functional characterization of CD107a perivascular progenitor cells will provide a foundation for future understanding of the role of these cells in developmental, regeneration and pathological process.

**Decision letter after peer review:**

Thank you for submitting your article "Lysosomal protein surface expression discriminates fat- from bone-forming human mesenchymal precursor cells" for consideration by *eLife*. Your article has been reviewed by three peer reviewers, and the evaluation has been overseen by a Reviewing Editor and Clifford Rosen as the Senior Editor. The following individual involved in review of your submission has agreed to reveal their identity: Yuji Mishina (Reviewer #2).

The reviewers have discussed the reviews with one another and the Reviewing Editor has drafted this decision to help you prepare a revised submission.

Summary:

In this study James et al. have performed extensive analysis of CD107a expression in ASCs and determined that CD107a^low^ expressing cells exhibit osteogenic potential both in vitro and in vivo. While all three reviewers are in agreement that the manuscript is well written and the establishment that CD107a serves as a surrogate marker to distinguish between adipogenic and osteogenic human ASCs would provide an important contribution to the field, they also have raised a number of questions (see below) that relate to the role of CD107a in cell fate. The reviewers and the editors realize the difficulty in performing the suggested additional in vivo bone regeneration experiments in a timely manner due to current Covid19 situation. However, the authors could use the in vitro model system to complement in vivo data to provide additional data to support the claims made in the paper. For example, the CD107a knockdown experiments using purified populations would provide more convincing data for direct involvement of CD107a in cell fate specification (or just an innocent bystander). Please consider clarifying some aspects noted below that were raised in the reviewer comments without additional experiments.

Please let us know if you have any questions.

Reviewer #1:

In this study, Peault B et al. claim that the cell surface expression of lysosomal associated protein-1 (LAMP1 or CD107a) serves as a surrogate marker to distinguish between osteogenic and adipogenic human adipose tissue derived stem cells (ASCs).

Using BD Lyoplate assay, authors identified CD107a as a cell surface protein differentially expressed between pericytes (CD146^+^CD34^-^) and adventicytes (CD146^-^CD34^+^). Authors recently demonstrated that these cells work together in bone formation (Wang et al., 2019). In this paper, authors show that CD107a^low^ ASC are more osteogenic relative to CD107^high^ cells, whereas the latter is more adipogenic than the former in vitro. in vitro experiments further demonstrate that CD107 cell surface expression is induced during adipogenesis. Gene silencing of CD107, however, does not interfere with adipogenic potential of ASCs. Consistent with in vitro findings, CD107a^high^ ASCs demonstrate highly osteogenic potential in ectopic bone formation and spinal fusion mouse models.

The potential significance of CD107a as a surrogate marker to enrich adipogenic ASCs is shown in vitro and the functional impact of CD107a^low^ ASCs in bone formation in vivo was well documented in this paper. However, the biological significance of lysosomal trafficking in ASCs is unclear.

1) Increased CD107a expression on the cell surface may simply reflect increased exocytosis process in differentiating preadipocytes. A subset of ASCs might be resistant to adipogenesis, which results in reduced CD107a expression on cell surface. From this study alone we do not know whether the intrinsic ability of lysosomal trafficking confers adipogenic versus osteogenic capacity of ASCs. Addressing the functional significance of lysosomal trafficking in adipogenesis may help us better understand the cellular mechanism of human ASC differentiation.

2) In vivo experiment of ectopic bone formation or bone fusion did not compare CD107a^low^ cells versus unsorted ASCs. If the focus of this paper is to use CD107a to enrich osteogenic ASCs for bone regeneration, comparing between enriched and unsorted cell populations will be needed.

3) Adipogenic potential of CD107a^high^ ASCs in vivo is not shown; as such, the in vivo significance of CD107a^high^ cells is unclear.

4) As discussed, developmental and pathological significance of CD107a in ASCs is unclear unless authors demonstrate the cell surface expression of this lysosomal protein in development and certain disease process, such as obesity.

5) Increased expression of CD107a in pericytes (CD146^+^) relative to adventicytes (CD34^+^) is intriguing. As the roles of pericytes versus fibroblasts in adipogenesis during development and in obesity are actively investigated at least in mice (1, 2), comparing the adipogenic potential of human pericytes relative to adventicytes may help us better understand human adipose tissue biology.

References:

1) Hepler C, Shan B, Zhang Q, Henry GH, Shao M, Vishvanath L, Ghaben AL, Mobley AB, Strand D, Hon GC, Gupta RK. Identification of functionally distinct fibro-inflammatory and adipogenic stromal subpopulations in visceral adipose tissue of adult mice. *eLife*. 2018;7:e39636. doi: 10.7554/*eLife*.39636.

2) Cattaneo P, Mukherjee D, Spinozzi S, Zhang L, Larcher V, Stallcup WB, Kataoka H, Chen J, Dimmeler S, Evans SM, Guimarães-Camboa N. Parallel Lineage-Tracing Studies Establish Fibroblasts as the Prevailing in vivo Adipocyte Progenitor. Cell Reports. 2020;30(2):571-82.e2. doi: https://doi.org/10.1016/j.celrep.2019.12.046.

Reviewer #2:

The authors identified endolysosomal protein CD107a as a novel cell surface antigen expressed among adipose tissue (AT)-resident perivascular stem cells. CD107a^low^ and CD107a^high^ AT-derived stromal cells represent osteoblast and adipocyte precursor cells, respectively. They also confirmed human CD107a^low^ cells drive dramatic bone formation using two mouse models. They demonstrated that CD107a correlates with exocytosis during early adipogenic differentiation, rather than having vital function in cellular differentiation. They further systematically compared the transcriptome of both cell types and identified CD107a^low^ cells are precursors of CD107a^high^ cells using bioinformatics analysis.

The authors did extensive and systemic analysis to compare CD107a^low^ and CD107a^high^ cell populations. The paper is well written and the identification of this new stem cell marker is an important contribution to the field.

This reviewer does not have any major issues.

Reviewer #3:

This is an interesting paper on the identification of CD107a as a novel marker of MSC to determine a subpopulation of the progenitors with specific differentiation and biological function in human stem cell biology. However, I have several comments that will require further attention:

1) In the Introduction, the authors mentioned the basic function of CD107a (Lamp1) marker as a lysosomal membrane protein. However, they did not mention its important role in autophagy, which should be included as it can make relevant explanation for their observation as the process of autophagy is crucial for regulation of adipogenesis (DOI: 10.1128/MCB.00193-13). The authors should add this information as it can be further discussed in relation to their findings.

2) Regarding the expression profile of CD107a in human adipose tissue, did the authors check the proportion of the cells and cell types with CD107a expression to identify the heterogeneity of the cells expressing this surface marker in AT and also if it correlates with number of lysosomes or their state of the activation?

3) Further, the authors mentioned that CD107a distinguish between AD and OB progenitors among MSC derived from AT. Did the authors check the expression profile of CD107a during AD and OB differentiation if its expression profile changes during these processes? They should add these data and comment on it in relation to CD107a function in MSC. Also, if there are known other factors e.g. inflammatory/metabolic molecules (e.g. TNFa, TGFb, or lipids) that could stimulate expression of CD107a?

4) Regarding the source of human AT samples, the authors should add the information on the age and BMI or possible medication used in the subjects as these are the major factors that can affect the quality and cellular composition of the tissue.

5) For the cell sorting, how stable was the expression of CD107a after sorting in the culture, did the CD107a^high^ and ^low^ cells maintain their difference in CD107a expression through the passaging? It is very common that MSC after sorting change their expression profile through the sub-culturing. The authors should add these data and comment on them.

6) Did the authors measure proliferation rate of CD107a^high^ and ^low^ expressed cells? Did they measure ALP activity or inflammatory properties (e.g. expression of cytokines, growth factors) in these cells? These are functional assays that would add more information about their cellular properties and determination.

7) Regarding the CFU-AD/OB data, how did the authors evaluate this parameter, i.e. they calculated only positive colonies or the ratio of positive colonies to total number of colonies? This should be specified to know if there was change in determination or switch in differentiation potential of the cells.

8) Regarding the different source of MSC on CD107a expression, the authors mentioned some data on CD107a expression in different fat depots or placenta (Discussion, fourth paragraph). They should add this information, also if they could add the data on BMSC and their expression of CD107a as these cells are more prone to osteogenesis. It will be interesting to compare different sources of MSC and the CD107a expression if it can be used for the specification of particular cell subpopulation.

9) Regarding the RNA-seq data between CD107a^low^ and ^high^ (Figure 4C), is the expression of AD genes significant between groups? It does not seem to be much different, e.g. *CEBPb*, *PPARg*, *Sirt1* indeed seem to be more expressed in CD107a^low^. The authors should more carefully interpret these data and correct their postulation.

10) Regarding the major outcome of this study, it is not clear what the authors would like to point out with their findings. Does it mean that MSC with lower CD107a expression have higher multipotent characteristics and better regeneration properties? They should elaborate more on these aspects of their findings and discuss it more with other literature. Also, if the testing regenerative properties of CD107a^low^ and ^high^ cells should be attributed, the authors should use better model for tissue regeneration for example calvarial defect regeneration model or monocortical bone defect model. As these models represent more appropriate system with injury stimuli for activation of the regenerative properties of the cells. The authors should work more on this part of the results.

---

## [Author Response]

Summary:In this study James et al. have performed extensive analysis of CD107a expression in ASCs and determined that CD107a^low^ expressing cells exhibit osteogenic potential both in vitro and in vivo. While all three reviewers are in agreement that the manuscript is well written and the establishment that CD107a serves as a surrogate marker to distinguish between adipogenic and osteogenic human ASCs would provide an important contribution to the field, they also have raised a number of questions (see below) that relate to the role of CD107a in cell fate. The reviewers and the editors realize the difficulty in performing the suggested additional in vivo bone regeneration experiments in a timely manner due to current Covid19 situation. However, the authors could use the in vitro model system to complement in vivo data to provide additional data to support the claims made in the paper. For example, the CD107a knockdown experiments using purified populations would provide more convincing data for direct involvement of CD107a in cell fate specification (or just an innocent bystander). Please consider clarifying some aspects noted below that were raised in the reviewer comments without additional experiments.

We thank the editor for this suggestion. To further pursue this, we examined CD107a knockdown in FACS isolated CD107a^low^ and CD107a^high^ cells. Our results showed that knockdown of CD107a in CD107a^low^ cells slightly promoted adipogenic differentiation, which was in similarity to unsorted cells. In CD107a^high^ cells, knockdown had no effect (revised Figure 3—figure supplement 4P, Q). We speculate that as CD107a^high^ cells have ^high^ intrinsic adipogenic potential, the effects of knockdown were not apparent in this group. In aggregate, these data suggest that CD107a is a marker (or innocent bystander), rather than directly involved in cell fate specification.

Reviewer #1:In this study, Peault B et al. claim that the cell surface expression of lysosomal associated protein-1 (LAMP1 or CD107a) serves as a surrogate marker to distinguish between osteogenic and adipogenic human adipose tissue derived stem cells (ASCs).Using BD Lyoplate assay, authors identified CD107a as a cell surface protein differentially expressed between pericytes (CD146^+^CD34^-^) and adventicytes (CD146^-^CD34^+^). Authors recently demonstrated that these cells work together in bone formation (1). In this paper, authors show that CD107a^low^ ASC are more osteogenic relative to CD107^high^ cells, whereas the latter is more adipogenic than the former in vitro. in vitro experiments further demonstrate that CD107 cell surface expression is induced during adipogenesis. Gene silencing of CD107, however, does not interfere with adipogenic potential of ASCs. Consistent with in vitro findings, CD107a^high^ ASCs demonstrate highly osteogenic potential in ectopic bone formation and spinal fusion mouse models.The potential significance of CD107a as a surrogate marker to enrich adipogenic ASCs is shown in vitro and the functional impact of CD107a^low^ ASCs in bone formation in vivo was well documented in this paper. However, the biological significance of lysosomal trafficking in ASCs is unclear.1) Increased CD107a expression on the cell surface may simply reflect increased exocytosis process in differentiating preadipocytes. A subset of ASCs might be resistant to adipogenesis, which results in reduced CD107a expression on cell surface. From this study alone we do not know whether the intrinsic ability of lysosomal trafficking confers adipogenic versus osteogenic capacity of ASCs. Addressing the functional significance of lysosomal trafficking in adipogenesis may help us better understand the cellular mechanism of human ASC differentiation.

We thank the reviewer for this suggestion. As you point out, it is likely that increased CD107a expression reflects increased exocytosis process in differentiating preadipocytes. In our past observations, CD107a knockdown led to a modest but significant increase in adipogenesis. To further expand on these results, adipogenic differentiation of ASCs was performed with Vac-1 treatment, the inhibitor of exocytosis which we previously observed prevented CD107a translocation to the cell membrane. Consistent with siRNA knockdown results, Vac-1 treatment also led to a modest increase in adipogenic differentiation (see Author response image 1). These results, along with new siRNA experiments using CD107a^low^ and ^high^ cells (Figure 3—figure supplement 4P, Q), suggest that partitioning allows for identification of functionally relevant subsets which is likely not explained by intrinsic function of CD107a protein. Changes to the Discussion to make this point more clear have been added (second paragraph).

**Author response image 1. sa2fig1:** Inhibition of exocytosis with Vacuolin-1 moderately promotes adipogenic differentiation of human ASCs. (A) Oil red O (ORO) staining (left) and photometric quantification (right) at d7 of adipogenic differentiation with or without Vacuolin-1 treatment (Vac-1, 1 μM). Black scale bar: 50 μm. (B-D) Adipogenic gene expression with or without Vac-1 (1 μM) treatment, including (B) Peroxisome proliferator-activated receptor-γ (PPARG), (C) Lipoprotein lipase (LPL), and (D) Fatty acid binding protein 4 (FABP4). Gene expression assessed at 7 d of adipogenic differentiation. Data repeated in experimental triplicate. *P<0.05; **P<0.01. Statistical analysis was performed using a two-tailed Student t-test.

2) In vivo experiment of ectopic bone formation or bone fusion did not compare CD107a^low^ cells versus unsorted ASCs. If the focus of this paper is to use CD107a to enrich osteogenic ASCs for bone regeneration, comparing between enriched and unsorted cell populations will be needed.

We thank the reviewer for the excellent suggestion. The main focus of the paper was to better understand the cellular heterogeneity of human progenitor cells within the perivascular niche. As pointed out by the editor, timely additional in vivo experiments are not currently feasible due to COVID19 related shutdowns in our area. However, we agree that the future use of CD107a cell subsets for tissue engineering efforts in the bone or fat field would require clear evidence of improvement in comparison to unsorted ASCs.

3) Adipogenic potential of CD107a^high^ ASCs in vivo is not shown; as such, the in vivo significance of CD107a^high^ cells is unclear.

Thank you for the excellent point. From our data and across 8 patient samples, we did find that CD107a^high^ stromal cells represent adipocyte precursor cells in vitro. Just as perivascular cells in PDGFRA^1^ or PDGFRB^2^ reporter animals have been observed to replenish adipocytes among either homeostasis or high fat diet conditions, it is quite possible that CD107a-expressing perivascular cells perform a similar function in vivo. Our studies are confined to human tissues, and future studies examining the dynamic cell fate of CD107a^high^ cells would need to be performed in the mouse in the context of lineage tracing. Of note, we did return to our muscle pouch and spine fusion assays and performed Oil red O staining to look for frank adipocytes within these tissues. No definite adipocytes were observed across any samples, suggesting that these particular carriers are not permissive to adipocyte formation. Future tissue engineering studies with CD107a^high^ cells for adipose tissue formation are indeed needed.

4) As discussed, developmental and pathological significance of CD107a in ASCs is unclear unless authors demonstrate the cell surface expression of this lysosomal protein in development and certain disease process, such as obesity.

Thank you for the good point. Mice deficient in either CD107a or CD107b are viable and fertile. However, mice deficient in both CD107a and CD107b have an embryonic lethal phenotype. In mouse embryonic fibroblasts, mutual disruption of both CD107s is associated with an increased accumulation of autophagic vacuoles and unesterified cholesterol^3^. In our preliminary data, CD107b is also expressed in the human adipose stromal cells. However, using FACS we found that only a small fraction of cells (0.39% in total ASCs) co-expressed surface CD107a and CD107b, making functional redundancy of these proteins in ASCs less likely. Any link between CD107a/CD107b and either obesity or a skeletal phenotype in mouse or humans is not known. In pilot studies, we did examine CD107a expression in a tissue microarray of healthy and diseased human coronary arteries. Although still preliminary data, adventitial and plaque associated CD107a immunostaining was present, particularly in diseased vessels (Author response image 2). It remains to be seen the extent to which CD107a-expressing adventitial cells are involved in pathologic vascular remodeling.

**Author response image 2. sa2fig2:** CD107a immunohistochemical staining in human atherosclerotic vessel. CD107a positive staining appears red. Red scale bar: 200 μm; black scale bar: 50 μm.

5) Increased expression of CD107a in pericytes (CD146^+^) relative to adventicytes (CD34^+^) is intriguing. As the roles of pericytes versus fibroblasts in adipogenesis during development and in obesity are actively investigated at least in mice (1, 2), comparing the adipogenic potential of human pericytes relative to adventicytes may help us better understand human adipose tissue biology.

Thank you for the interesting point. In our experience, adventitial cells rather than pericytes have a higher intrinsic adipogenic potential, which most likely correlates with their theoretically higher ‘stem-like’ identity^4^. In terms of frequency, and across 8 patient samples, membranous CD107a was present in CD146^+^ pericytes, CD34^+^ adventicytes and also a fraction of as of yet undefined mesenchymal cells (C146^-^CD34^-^CD31^-^CD45^-^ cells). Interestingly, the percentage of pericytes and adventicytes that made up the CD107a^high^ cell population varied to some degree across samples (see Supplementary files 2 and 3). In the future, membranous CD107a among mesenchymal cells or a subfraction of perivascular mesenchymal cells could be used prospectively as a biomarker. Nevertheless, our study was a first step in analyzing a completely unknown marker in MSC biology. The further functional segregation of CD107a-expressing cell subsets (pericytes and adventitial cells) is a fascinating area of research, but lies considerably outside the scope of the present work.

References:1) Hepler C, Shan B, Zhang Q, Henry GH, Shao M, Vishvanath L, Ghaben AL, Mobley AB, Strand D, Hon GC, Gupta RK. Identification of functionally distinct fibro-inflammatory and adipogenic stromal subpopulations in visceral adipose tissue of adult mice. eLife. 2018;7:e39636. doi: 10.7554/eLife.39636.2) Cattaneo P, Mukherjee D, Spinozzi S, Zhang L, Larcher V, Stallcup WB, Kataoka H, Chen J, Dimmeler S, Evans SM, Guimarães-Camboa N. Parallel Lineage-Tracing Studies Establish Fibroblasts as the Prevailing in vivo Adipocyte Progenitor. Cell Reports. 2020;30(2):571-82.e2. doi: https://doi.org/10.1016/j.celrep.2019.12.046.Reviewer #3:This is an interesting paper on the identification of CD107a as a novel marker of MSC to determine a subpopulation of the progenitors with specific differentiation and biological function in human stem cell biology. However, I have several comments that will require further attention:1) In the Introduction, the authors mentioned the basic function of CD107a (Lamp1) marker as a lysosomal membrane protein. However, they did not mention its important role in autophagy, which should be included as it can make relevant explanation for their observation as the process of autophagy is crucial for regulation of adipogenesis (DOI: 10.1128/MCB.00193-13). The authors should add this information as it can be further discussed in relation to their findings.

Thank you for the excellent point, which we have added to the Discussion (second paragraph). CD107a is responsible for maintaining lysosomal integrity, pH and catabolism^3,5^. Lysosomes provide the degradative enzymes for autophagy and participate in autophagy regulation^6^, and CD107a may have an essential role in the process of autophagy. Also as you point out, the activation of autophagy is crucial for adipogenic differentiation^7^. Whether CD107a^high^ cells enhance adipogenesis by promoting autophagy is an interesting and worthy follow-up topic. This additional discussion has been added to the resubmission (see the aforementioned paragraph).

2) Regarding the expression profile of CD107a in human adipose tissue, did the authors check the proportion of the cells and cell types with CD107a expression to identify the heterogeneity of the cells expressing this surface marker in AT and also if it correlates with number of lysosomes or their state of the activation?

Thank you for the excellent suggestion. The proportion of cell types present within CD107a^high^ and ^low^ cell preparations is reported in Supplementary files 2 and 3. Briefly, both CD107a^high^ and ^low^ cell populations showed relatively equivalent numbers of CD146^+^ pericytes (6.94% in CD107a^low^ cells and 4.47% in CD107a^high^ cells), while CD107a^low^ cells contained a slightly greater population of CD34^+^ adventitial cells (70.88% in CD107a^low^ cells and 52.45% in CD107a^high^ cells). According to your suggestion, we quantified the number of lysosomes among the CD107a^low^ and CD107a^high^ cells using Lysosomal Staining Kit (Abcam) and found no difference in lysosome staining intensity (see Author response image 3).

**Author response image 3. sa2fig3:** No change in lysosome immunohistochemical staining among human CD107a^low^ and CD107a^high^ cells. Lysosome positive staining appears green. White scale bar: 20 μm.

3) Further, the authors mentioned that CD107a distinguish between AD and OB progenitors among MSC derived from AT. Did the authors check the expression profile of CD107a during AD and OB differentiation if its expression profile changes during these processes? They should add these data and comment on it in relation to CD107a function in MSC. Also, if there are known other factors e.g. inflammatory/metabolic molecules (e.g. TNFa, TGFb, or lipids) that could stimulate expression of CD107a?

Thank you for the excellent suggestions. We observed that CD107a was increased prominently during early adipogenic differentiation (Figure 3H-J). Briefly, after 3 d exposure to adipogenic conditions, a 12.03-fold increase in immunostaining intensity and a 253.5% increase in the number of CD107a^high^ cells were noted. In contrast, no significant change in membranous CD107a expression profile was found during osteogenic differentiation (Figure 3—figure supplement 2). As to your good point regarding inflammatory mediators of CD107a expression, this is well reported in other cell types. For example, IL-2 stimulates CD107a expression on NK and cytotoxic T cells^8^. In addition, CD107a production is increased with the stimulation of IL-15, IFN-α, and IFN-β in NK cells^9^. Linking perivascular inflammation to mesenchymal CD107a expression in obesity or other pathophysiology would be an interesting subject for the future.

4) Regarding the source of human AT samples, the authors should add the information on the age and BMI or possible medication used in the subjects as these are the major factors that can affect the quality and cellular composition of the tissue.

Thank you for the suggestions. Details of age and BMI have been added in Supplementary file 1. Unfortunately, data on medication usage was not available for these de-identified samples.

5) For the cell sorting, how stable was the expression of CD107a after sorting in the culture, did the CD107a^high^ and ^low^ cells maintain their difference in CD107a expression through the passaging? It is very common that MSC after sorting change their expression profile through the sub-culturing. The authors should add these data and comment on them.

Thank you for the good point. The frequency of membranous CD107a in CD107a^high^ cells did not change after three passages. However, the frequency of membranous CD107a among passaged CD107a^low^ cells increased slightly over three passages (1.41%).

6) Did the authors measure proliferation rate of CD107a^high^ and ^low^ expressed cells? Did they measure ALP activity or inflammatory properties (e.g. expression of cytokines, growth factors) in these cells? These are functional assays that would add more information about their cellular properties and determination.

Thank you for the suggestion. Proliferation assays have been added, in which we observed that CD107a^low^ cells have a higher proliferative index in comparison to CD107a^high^ cells (Figure 2A, subsection “CD107a^low^ AT-derived stromal cells represent osteoblast precursor cells!). As suggested, we observed that ALP activity was quite different among CD107a^low^ and ^high^ cells (Figure 2I, J), in which ALP staining and quantification demonstrated an enrichment among CD107a^low^ cells. According to your suggestions, we re-analyzed existing RNA sequencing data and found that CD107a^low^ and ^high^ cells expressed few inflammatory cytokines, such as *IL-1B*, *IL-6*, and *IL-33*. In addition, both cell populations produced common growth factors such as *FGF2, BMP2, PDGFA*, *PDGFB, VEGFA* and *IGF1*, and at comparable levels (see Author response image 4).

**Author response image 4. sa2fig4:** Expression of inflammatory cytokines and common growth and differentiation factors among human CD107a^low^CD31^-^CD45^-^ and CD107a^high^CD31^-^CD45^-^ mesenchymal cells. Data shown in volcano plot. X-axis represents Log2 fold change for each gene. Y-axis represents -Log10 p value. Red dots indicate >2SD increase among CD107a^high^ mesenchymal cells. Blue dots indicate >2SD increase among CD107a^low^ mesenchymal cells. Inflammatory cytokines are shown in purple box, while growth factors are shown in green box.

7) Regarding the CFU-AD/OB data, how did the authors evaluate this parameter, i.e. they calculated only positive colonies or the ratio of positive colonies to total number of colonies? This should be specified to know if there was change in determination or switch in differentiation potential of the cells.

Thank you for the good point. For CFU-OB/AD quantification, we calculated the total number of positive colonies in each well. This has been added in the Materials and methods section (subsection “Colony-forming unit (CFU) assay”).

8) Regarding the different source of MSC on CD107a expression, the authors mentioned some data on CD107a expression in different fat depots or placenta (Discussion, fourth paragraph). They should add this information, also if they could add the data on BMSC and their expression of CD107a as these cells are more prone to osteogenesis. It will be interesting to compare different sources of MSC and the CD107a expression if it can be used for the specification of particular cell subpopulation.

Thank you for the suggestion. CD107a expression within different fat depots is shown in Figure 1—figure supplement 2, in which CD107a immunoreactivity within the perivascular mesenchymal niche was noted, including pericardial, perigonadal, perirenal and omental human fat. In addition, we have begun to investigate disparate tissues such as placenta and fetal skeletal muscle. An example of the perivascular staining patterns are shown in Author response image 5. However, this distinct project is being performed by another graduate student, and is reserved for a distinct thesis project. Finally, we did observe that a portion of BMSC expressed membranous CD107a (2.54%, Figure 3L). Interestingly, BMSC showed a similar increase in CD107a expression during adipogenesis as did ASC (Figure 3L). We hope in future work to more thoroughly understand how CD107a identifies perivascular cell subsets across and between tissues.

**Author response image 5. sa2fig5:** CD107a immunohistochemical staining in human fetal skeletal muscle. CD107a positive staining appears red, while CD31 positive staining appears black.

9) Regarding the RNA-seq data between CD107a^low^ and ^high^ (Figure 4C), is the expression of AD genes significant between groups? It does not seem to be much different, e.g. CEBPb, PPARg, Sirt1 indeed seem to be more expressed in CD107a^low^. The authors should more carefully interpret these data and correct their postulation.

Thank you for the suggestion. *LRP5* (3.76-fold increase; p value 0.0028), *FABP4* (1.40-fold increase; p value 0.0160), *NCOR2* (4.69-fold increase; p value 0.0060), and *ANGPT2* (2.26-fold increase; p value 0.0220) were significantly increased in CD107a^high^ stromal cells. *CEBPA* (1.08-fold increase; p value 0.8488), *CEBPD* (1.48-fold increase; p value 0.0873), and *LPL* (1.11-fold increase; p value 0.3596) trended towards an upregulation among in CD107a^high^ cells, but this did not reach statistical significance. Some significantly upregulated genes in CD107a^low^ stromal cells were negative regulators of adipogenesis, such as *KLF2*, *KLF3*, *SIRT1*, and *DDIT3*^10-13^ (subsection “Transcriptomic analysis suggests a progenitor cell phenotype for CD107a^low^ cells”, first paragraph).

10) Regarding the major outcome of this study, it is not clear what the authors would like to point out with their findings. Does it mean that MSC with lower CD107a expression have higher multipotent characteristics and better regeneration properties? They should elaborate more on these aspects of their findings and discuss it more with other literature. Also, if the testing regenerative properties of CD107a^low^ and ^high^ cells should be attributed, the authors should use better model for tissue regeneration for example calvarial defect regeneration model or monocortical bone defect model. As these models represent more appropriate system with injury stimuli for activation of the regenerative properties of the cells. The authors should work more on this part of the results.

Thank you for the excellent suggestions and the opportunity to summarize. Our main findings were that CD107a^low^ cells demonstrate high colony formation, osteoprogenitor cell frequency, and osteogenic potential in vitro and in vivo. Conversely, CD107a^high^ cells include almost exclusively adipocyte progenitor cells and exhibit high fat cell forming potential. Overall this suggests a more ‘stem-like’ identity for CD107a^low^ cells, and that CD107a^high^ cell populations have a high pre-adipocyte content which is functionally associated with exocytosis during early adipogenic differentiation.

According to your earlier suggestion, we have reviewed in more detail the literature regarding CD107a regulation of autophagy (Discussion, second paragraph), as well as inflammatory inducers of CD107a expression (see response to point 3). As you point out, there are many potential bone injury models that could be chosen, including monocortical bone defects or calvarial bone defects. In our experience with the mentioned models, these models rely on both paracrine stimulation of endogenous cells and direct ossification of the implanted cells. In contrast, the spine fusion model relies more on direct ossification of the implanted cells, which was one of the primary factors which influenced our choice of models.

References:

1) Wang Y, Xu J, Meyers CA, et al. PDGFRalpha marks distinct perivascular populations with different osteogenic potential within adipose tissue. Stem Cells 38, 276-290, doi:10.1002/stem.3108 (2020).

2) Gao Z, Daquinag, AC, Su F, Snyder B and Kolonin MG. PDGFRalpha/PDGFRbeta signaling balance modulates progenitor cell differentiation into white and beige adipocytes. Development 145, doi:10.1242/dev.155861 (2018).

3) Eeva-Liisa E. Roles of LAMP-1 and LAMP-2 in lysosome biogenesis and autophagy. Mol Aspects Med 27(5-6), 495-502, doi: 10.1016/j.mam.2006.08.005 (2006).

4) Hardy WR, Moldovan N, Moldovan L, et al. Transcriptional Networks in Single Perivascular Cells Sorted from Human Adipose Tissue Reveal a Hierarchy of Mesenchymal Stem Cells. Stem Cells 35(5), 1273-1289, doi: 10.1002/stem.2599 (2017).

5) Andrejewski N, Punnonen EL, Guhde G, et al. Normal lysosomal morphology and function in LAMP-1-deficient mice. The Journal of Biological Chemistry 274(18), 12692–12701, doi: 10.1074/jbc.274.18.12692 (1999).

6) Yim WW, Mizushima N. Lysosome biology in autophagy. Cell Discovery 6, doi: 10.1038/s41421-020-0141-7 (2020).

7). Guo L, Huang J, Liu Y, et al. Transactivation of Atg4b by C/EBPβ promotes autophagy to facilitate adipogenesis. Mol Cell Biol 33(16), 3180-3190, doi: 10.1128/MCB.00193-13 (2013).

8) Aktas E, Kucuksezer UC, Bilgic S, et al. Relationship between CD107a expression and cytotoxic activity. Cellular Immunology 254(2), 149-154, doi: 10.1016/j.cellimm.2008.08.007 (2009).

9) Xie Z, Zheng J, Wang Y, et al. Deficient IL-2 produced by activated CD56+ T cells contributes to impaired NK cell-mediated ADCC function in chronic HIV-1 infection. Front Immunol 10, 1647, doi: 10.3389/fimmu.2019.01647 (2019).

10) Banerjee S, Feinberg M, Watanabe M, et al. The Krüppel-like factor *KLF2* inhibits peroxisome proliferator-activated receptor-γ expression and adipogenesis. J Biol Chem 278(4), 2581-2584, doi: 10.1074/jbc.M210859200 (2003).

11) Sue N, Jack B, Eaton S, et al. Targeted disruption of the basic Krüppel-like factor gene (*Klf3*) reveals a role in adipogenesis. Mol Cell Biol 28(12), 3967-3978, doi: 10.1128/MCB.01942-07 (2008).

12) Zhou Y, Zhou Z, Zhang W, et al. *SIRT1* inhibits adipogenesis and promotes myogenic differentiation in C3H10T1/2 pluripotent cells by regulating Wnt signaling. Cell Biosci 5, 61, doi: 10.1186/s13578-015-0055-5 (2015).

13) Pereira R, Delany A, Canalis E. CCAAT/enhancer binding protein homologous protein (*DDIT3*) induces osteoblastic cell differentiation. Endocrinology 145(4), 1952-1960, doi: 10.1210/en.2003-0868 (2004).